# The Curse of Low Task Diversity: On the Failure of Transfer Learning to Outperform MAML and their Empirical Equivalence

## Abstract

Recently, it has been observed that a transfer learning solution might be all we need to solve many few-shot learning benchmarks – thus raising important questions about when and how meta-learning algorithms should be deployed. In this paper, we seek to clarify these questions by 1. proposing a novel metric – the *diversity coefficient* – to measure the diversity of tasks in a few-shot learning benchmark and 2. by comparing MAML and transfer learning under fair conditions (same architecture, same optimizer and all models trained to convergence). Using the diversity coefficient, we show that the popular MiniImagenet and Cifar-fs few-shot learning benchmarks have low diversity. This novel insight contextualizes claims that transfer learning solutions are better than meta-learned solutions in the regime of low diversity under a fair comparison. Specifically, we empirically find that a low diversity coefficient correlates with a high similarity between transfer learning and Model-Agnostic Meta-Learning (MAML) learned solutions in terms of accuracy at meta-test time and classification layer similarity (using feature based distance metrics like SVCCA, PWCCA, CKA, and OPD). To further support our claim, we find this meta-test accuracy holds even as the model size changes. Therefore, we conclude that in the low diversity regime, MAML and transfer learning have equivalent meta-test performance when both are compared fairly. We also hope our work inspires more thoughtful constructions and quantitative evaluations of meta-learning benchmarks in the future.

## 1 Introduction

The success of deep learning in computer vision (1; 2), natural language processing (3; 4), game playing (5; 6; 7) and more, keeps motivating a growing body of applications of deep learning on an increasingly wide variety of domains. In particular, deep learning is now routinely applied to few-shot learning – a research challenge that assesses a model's ability to learn to adapt to new tasks, new distributions, or new environments. This has been the main research area where meta-learning algorithms have been applied – since such a strategy seems promising in a small data regime due to its potential to *learn to learn* or *learn to adapt*. However, it was recently shown (8) that a transfer learning model with a fixed embedding can match and outperform many modern sophisticated meta-learning algorithms on numerous few-shot learning benchmarks (9; 10; 11; 12). This growing body of evidence – coupled with these surprising results in meta-learning – raise the question if researchers are applying meta-learning with the right inductive biases (13; 14) and designing appropriate benchmarks for meta-learning. Our evidence suggests this is not the case.

In this work, we show that when the task diversity – a novel measure of variability across tasks – is low, then MAML (Model Agnostic Meta-Learning) (15) learned solutions have the same accuracy

Submitted to 36th Conference on Neural Information Processing Systems (NeurIPS 2022). Do not distribute.

as transfer learning (i.e., a supervised learned model with a fine-tuned final linear layer). We want to emphasize the importance of doing such an analysis fairly: with the same architecture, same optimizer and all models trained to convergence. This empirical equivalence remained true even as the model size changed – thus further suggesting this equivalence is more a property of the data than of the model. Therefore, we suggest taking a problem-centric approach to meta-learning and suggest applying Marr's level of analysis ([16; 17]) to few-shot learning – to identify the family of problems suitable for meta-learning. Marr emphasized the importance of understanding the computational problem being solved and not only analyzing the algorithms or hardware that attempts to solve them. An example given by Marr is marveling at the rich structure of bird feathers without also understanding the problem they solve is flight. Similarly, there has been analysis of MAML solutions and transfer learning without putting the problem such solutions should solve into perspective ([18; 19]). Therefore, in this work, we hope to clarify some of these results by partially placing the current state of affairs in meta-learning from a problem-centric view. In addition, the novelty of our analysis compared to previous work is that we make analysis intrinsic of the data as a first class citizen.

**Our contributions** summarized as follows:

1. *We propose a novel metric that quantifies the **intrinsic diversity** of the data of a few-shot learning benchmark.* We call it the *diversity coefficient*. It enables analysis of meta-learning algorithms through a *problem-centric framework*. It also goes beyond counting the number of classes or number of data points or counting the number concatenated data sets – and instead quantifies the expected diversity/variability of tasks in a few-shot learning benchmark.

2. *We analyze the two most prominent few-shot learning benchmarks – MiniImagenet and Cifar-fs – and show that their diversity is low.* These results are robust across different ways to measure the diversity coefficient, suggesting that our approach is robust.

3. With this context, we partially clarify the surprising results from ([19]) by comparing their transfer learning method against models trained with MAML ([15]). *In particular, when making a fair comparison, transfer learning method with a fixed feature extractor fails to outperform MAML.* We define a fair comparison when the two methods are compared using the same architecture (backbone), same optimizer and all models trained to convergence. We also show that their final layer makes similar predictions according to neural network distance techniques like distance based Singular Value Canonical Correlation Analysis (SVCCA), Projection Weighted (PWCCA), Linear Centered Kernel Analysis (LINCKA) and Orthogonal Procrustes Distance (OPD). This equivalence holds even as the model size increases.

4. Interestingly, we also find that even in the regime where task diversity is low (in MiniImagenet and Cifar-fs), the features extracted by supervised learning and MAML are different – implying that the mechanism by which they function is different despite the similarity of their final predictions.

5. *As an actionable conclusion, we provide a metric that can be used to analyze the intrinsic diversity of the data in a few-shot learning benchmarks and therefore build more thoughtful environments to drive research in meta-learning.* In addition, our evidence suggests the following test to predict the empirical equivalence of MAML and transfer learning: if the task diversity is low, then transfer learned solutions might fail to outperform meta-learned solutions. This test is easy to run because our diversity coefficient can be done using the Task2Vec method ([20]) using pre-trained neural network. We also found that random networks were consistent with the results of pre-trained networks on Imagenet.

We hope that this line of work inspires a problem-centric first approach to meta-learning – which appears to be especially sensitive to the properties of the problem in question. Therefore, we hope future work takes a more thoughtful and **quantitative** approach to benchmark creation – instead of focusing only on making huge data sets.

## 2 Background

In this section, we provide a summary of the background needed to understand our main results.

**Model-Agnostic Meta-Learning (MAML):** The MAML algorithm ([15]) attempts to meta-learn an initialization of parameters for a neural network so that it is primed for fast gradient descent

adaptation. It consists of two main optimization loops: 1) an outer loop used to prime the parameters for fast adaptation, and 2) an inner loop that does the fast adaptation. During meta-testing, only the inner loop is used to adapt the representation learned by the outer loop.

**Transfer Learning with Union Supervised Learning (USL):** Previous work [19] shows that an initialization trained with supervised learning, on a union of all tasks, can outperform many sophisticated methods in meta-learning. In particular, their method consists of two stages: 1) first they use a union of all the labels in the few-shot learning benchmark during meta-training and train with standard supervised learning (SL), then 2) during the meta-testing, they use an inference method common in transfer learning: extract a fixed feature from the neural network and fully fine-tune the final classification layer (i.e., the head). Note that our experiments only consider when the final layer is regularized Logistic Regression trained with LBGFS.

**Distances for Deep Neural Network Feature Analysis:** To compute the distance between neural networks we use the distance versions of Singular Value Canonical Correlation Analysis (SVCCA) [21], Projection Weighted Canonical Correlation (PWCCA) [22], Linear Centered Kernel Analysis (LINCKA) [23] and Orthogonal Procrustes Distance (OPD) [24]. These distances are in the interval $[0, 1]$ and are not necessarily a formal distance metric but are guaranteed to be zero when their inputs are equal and nonzero otherwise. This is true because SVCCA, PWCCA, LINCKA are based on similarity metrics and OPD is already a distance. Note that we use the formula $d(X, Y) = 1 - sim(X, Y)$ for our distance metrics where $sim$ is one either SVCCA, PWCCA, LINCKA similarity metric and $X, Y$ are matrices of activations (called layer matrices). The distance between two models is computed by choosing a layer and then comparing the features/activations after adaptation for that layer given a batch of tasks represented as a support and query set. A more thorough overview of these metrics for the analysis of internal representations for convolutional neural networks (CNNS) can be found in the appendix, section G.

**Task2Vec Embeddings for Distances between Tasks:** The diversity coefficient we propose is the expectation of distance between tasks (explain in more detail in section 3). Therefore, it is essential to define the distance between different pairs of tasks. We choose the cosine distance between Task2Vec (vectorial) embeddings as in [20]. Therefore, we provide a summary of the Task2Vec method to compute task embeddings. The vectorial representation of tasks provided by Task2Vec [20] is the vector of diagonal entries of the Fisher Information Matrix (FIM) given a fix neural network as a feature extractor – also called a **probe network** – after fine-tuning the final classification layer to the task. The authors explain this is a good vectorial representation of tasks because 1. It approximately indicates the most informative weights for solving the current task (up to a second order approximation) 2. For rich probe networks like CNNs, the diagonal is more computationally tractable. We choose Task2Vec because the original authors provide extensive evidence that their embeddings correlate with semantic and taxonomic relations between different visual classes – making it a convincing embedding for tasks [20]. The Task2Vec embedding of task $\tau$ is the diagonal of the following matrix:

$$\hat{F}_{D_\tau, f_w} = \hat{F}(D_\tau, f_w) = \mathbb{E}_{x,y \sim \hat{p}(x|\tau)p(y|x,f_w)}[\nabla_w \log p(y \mid x, f_w) \nabla_w p(y \mid x, f_w)^\top] \tag{1}$$

where $f_w$ is the neural networks used as a feature extractor with architecture $f$ and weights $w$, $\hat{p}(x \mid \tau)$ is the empirical distribution defined by the training data $D_\tau = \{(x_i, y_i)\}_{i=1}^n$ for task $\tau$, and $p(y \mid x, f_w)$ is a deep neural network trained to approximate the (empirical) posterior $\hat{p}(y \mid x, \tau)$. We'd like to emphasize that the there is a dependence on target label since Task2Vec fixes the feature extractor (using $f_w$) and then fits the final layer (or "head") to approximate the task posterior distribution $\hat{p}(y \mid x, \tau)$.

## 3   Definition of the Diversity Coefficient

The diversity coefficient aims to measure the intrinsic diversity (or variability) of tasks in a few-shot learning benchmark. At a high level, the diversity coefficient is the expected distance between a pair of different tasks **given a fixed probe network**. In this work, we choose the distance to be the cosine distance between vectorial representations (i.e. embeddings) of tasks according to Task2Vec as described in section 2. Using a fixed probe networks is essential because: 1. Using a fixed probe network means that the distances between different tasks are comparable, as discussed in the original Task2Vec [20] and 2. Since we are computing the distance between different tasks, we need to make sure the difference comes from intrinsic properties of the data and not from a different source, e.g. if

142 one uses different models then this might confound the source of variability in our metric. We define
143 the **diversity coefficient** of a few-shot learning benchmark $B$ as follows:

$$\hat{div}(B) = \mathbb{E}_{\tau_1 \sim \hat{p}(\tau|B), \tau_2 \sim \hat{p}(\tau|B)} \mathbb{E}_{D_1 \sim \hat{p}(x_1, y_1|\tau_1), D_2 \sim \hat{p}(x_2, y_2|\tau_2)} \left[ d(\hat{F}_{D_1, f_w}, \hat{F}_{D_2, f_w}) \right] \quad (2)$$

144 where $f_w$ is the neural networks used as a feature extractor with architecture $f$ and weights $w$,
145 $\hat{p}(x \mid \tau)$ is the empirical distribution defined by the training data $D_\tau = \{(x_i, y_i)\}_{i=1}^n$ for task $\tau$,
146 $\tau_1, \tau_2$ are tasks sampled from the empirical distribution of tasks $\hat{p}(\tau \mid B)$ for the current benchmark
147 $B$ (i.e. a batch of tasks with their data sets $\mathcal{D} = (\tau_i, D_{\tau_i})_{i=1}^N$), a task $\tau_i$ is the probability distribution
148 $p(x, y \mid \tau)$ of the data, $d$ is a distance metric (for us cosine), $f_w$ is the neural networks used as
149 a feature extractor with architecture $f$ and weights $w$, and $\hat{p}(x \mid \tau)$ is the empirical distribution
150 defined by the training data $D_\tau = \{(x_i, y_i)\}_{i=1}^n$ for task $\tau$. We'd also like to recall the reader that the
151 definition of a task in this setting is of a n-way, k-shot few-shot learning task. Therefore, each task has
152 n classes sampled with k examples used for the adaptation. We'd like to emphasize that the adaptation
153 here is only to fine-tune the final layer according to the Task2Vec method for the correct computation
154 of the FIM. Therefore, in this setting we combine the support and query set as the split is not relevant
155 for the computation of the task embedding using Task2Vec. Note that the above formulation can be
156 easily adapted to any distance function between tasks, and is not necessarily specific to using the
157 FIM or cosine distance. For example, given the true distributions for tasks one can use real distances
158 between probability distributions e.g. Hellinger distance. In addition, it is obvious one can use a
159 distance function besides the cosine distance – but choose it in accordance to the original work of
160 Task2Vec [20].

# 4 Experiments

162 This section explains the experiments backing up our main results outlined in our list of contributions.
163 Experimental details are provided in the supplementary section A and the learning curves displaying
164 the convergence for a fair comparison are in supplementary section B.

## 4.1 The Diversity Coefficient of MiniImagenet and Cifar-fs

166 To put our analysis into a problem-centric framework, we first analyze the problem they are trying
167 to solve through the diversity coefficient. Recall that the diversity coefficient aims to quantify the
168 intrinsic variation of tasks in a few-shot learning benchmark. We show that the diversity coefficient
169 of the popular MiniImagenet and Cifar-fs benchmarks are low with good confidence intervals using
170 four different probe networks in table 1.

| Probe Network | Diversity on MI | Diversity on Cifar-fs |
|---|---|---|
| Resnet18 (pt) | $0.117 \pm 2.098e\text{-}5$ | $0.100 \pm 2.18e\text{-}5$ |
| Resnet18 (rand) | $0.0955 \pm 1.29e\text{-}5$ | $0.103 \pm 1.05e\text{-}5$ |
| Resnet34 (pt) | $0.0999 \pm 1.95e\text{-}5$ | $0.0847 \pm 3.06e\text{-}5$ |
| Resnet34 (rand) | $0.0620 \pm 8.12e\text{-}6$ | $0.0643 \pm 9.64e\text{-}6$ |

Table 1: **The diversity coefficient of MiniImagenet (MI) and Cifar-fs is low.** The diversity coefficient was computed using the cosine distance between different standard n-way, k-shot classification tasks from the few-shot learning benchmark using the Task2Vec method described in section 3. We used n=5 (number of classes) and k=20 (number of examples per class) since we can use the whole task data to compute the diversity coefficient (no splitting of support and query set required). We used Resnet18 and Resnet34 networks as probe networks – both pre-trained on ImageNet (indicated as "pt" on table) and randomly initialized (indicated as "rand" on table). We observe that both type of networks and weights give similar diversity results. All confidence intervals were at 95%. To compute results, we used 500 few-shot learning tasks and only compare pairs of different tasks. This results in $(500^2 - 500)/2 = 124,750$ pair-wise distances used to compute the diversity coefficient.

## 4.2 Low Diversity Correlates with Equivalence of MAML and Transfer Learning

172 Now that we have placed ourselves in a problem-centric framework and shown the diversity coefficient
173 of the popular MiniImagenet and Cifar-fs benchmarks are low – we proceed to show the failure of

transfer learning (with USL) to outperform MAML. Crucially, the analysis was done using a fair comparison: using the same model architecture, optimizer, and training all models to convergence – details in section A. We used the five-layer CNN used in (15; 25) and Resnet12 as in (19). We provide evidence that in the setting of low diversity:

1. The accuracy of an adapted MAML meta-learner vs. an adapted USL pre-trained model are similar and statistically significant, except for one result where transfer learning with USL is worse. This is shown in table 2 and 1.

2. The distance for the classification layer decreases sharply according to four distance-based metrics – SVCCA, PWCCA, LINCKA, and OPD – as shown in figure 2. This implies the predictions of the two are similar.

For the first point, we emphasize that tables 1 and table 2 taken together support our central hypothesis: that models trained with meta-learning are not inferior to transfer learning models (using USL) when the diversity coefficient is low. Careful inspection reveals that the methods have the same meta-test accuracy with intersecting confidence intervals – making the results statistically significant across few-shot benchmarks and architectures. The one exception is the third set of bar plots, where transfer learning with USL is in fact worse.

For the second point, refer to figure 2 and observe that as the depth of the network increases, the distance between the activation layers of a model trained with MAML vs USL increases until it reaches the final classification layer – where all four metrics display a noticeable dip. In particular, PWCCA considers the two prediction layers identical (approximately zero distance). This final point is particularly interesting because PWCCA is weighted according to the CCA weights that stabilize with the final predictions of the network. This means that the PWCCA distance value is reflective of what the networked actually learned and gives a more reliable distance metric (for details, refer to the appendix section G.5). This is important because this supports our main hypothesis: that at prediction time there is an equivalence between transfer learning and MAML when the diversity coefficient is low.

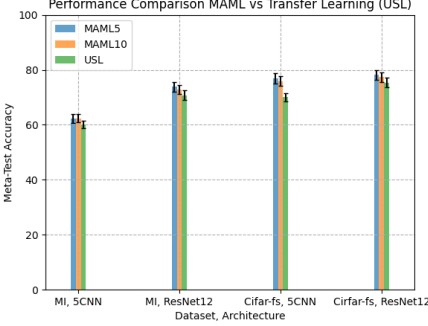

Figure 1: **MAML trained models and union supervised trained (USL) models have statistically equivalent meta-test accuracy for MiniImagenet and Cifar-fs with Resnet12 and five layer CNNs.** This holds for both the Resnet12 architecture used in (19) and the 5 layer CNN (indicated as "5CNN") in (25). Results used a (meta) batch-size of 100 tasks and 95% confidence intervals. All MAML models were trained with 5 inner steps during meta-training. "MAML5" and "MAML10" in the bar plot indicates the adaptation method used at test time i.e. we used 5 inner steps and 10 inner steps at test time. MiniImagenet is abbreviated as "MI" in the figure.

## 4.3 Is the Equivalence of MAML and Transfer Learning related to Model Size or Low Diversity?

An alternative hypothesis to explain the equivalence of transfer learning (with USL) and MAML could be due to the capabilities of large neural networks to be better meta-learners in general. Inspired by the impressive ability of large language models to be few-shot (or even zero-shot) learners (4; 27; 28; 3) – we hypothesized that perhaps the meta-learning capabilities of deep learning models is a function of the model size. If this were true, then we expected to see the difference in meta-test accuracy

| Meta-train Initialization | Adaptation at Inference | Meta-test Accuracy |
|---|---|---|
| Random | no adaptation | $19.3 \pm 0.80$ |
| MAML0 | no adaptation | $20.0 \pm 0.00$ |
| USL | no adaptation | $15.0 \pm 0.26$ |
| Random | MAML5 adaptation | $34.2 \pm 1.16$ |
| **MAML5** | **MAML5 adaptation** | **$62.4 \pm 1.64$** |
| USL | MAML5 adaptation | $25.1 \pm 0.98$ |
| Random | MAML10 adaptation | $34.1 \pm 1.23$ |
| **MAML5** | **MAML10 adaptation** | **$62.3 \pm 1.50$** |
| USL | MAML10 adaptation | $25.1 \pm 0.97$ |
| Random | Adapt Head only (with LR) | $40.2 \pm 1.30$ |
| MAML5 | Adapt Head only (with LR) | $59.7 \pm 1.37$ |
| **USL** | **Adapt Head only (with LR)** | **$60.1 \pm 1.37$** |

Table 2: **MAML trained representations and supervised trained representation have statistically equivalent meta-test accuracy on MiniImagenet – which has low diversity.** The transfer model's adaptation is labeled as "Adapted Head only (with LR)" – which stands for "Logistic Regression (LR)" used in [19]. More precisely, we used Logistic Regression (LR) with LBFGS with the default value for the l2 regularization parameter given by Python's Sklearn. Note that an increase in inner steps from 5 to 10 with the MAML5 trained model does not provide an additional meta-test accuracy boost, consistent with previous work [26]. Note that the fact that the MAML5 representation matches the USL representation when both use the same adaptation method is not surprising – given that: 1) previous work has shown that the distance between the body of an adapted MAML model is minimal compared to the unadapted MAML (which we reproduce in 5 in the green line) and 2) the fact that a MAML5 adaptation is only 5 steps of MAML while LR fully converges the prediction layer. We want to highlight that only the MAML5 model achieved the maximum meta-test performance of 0.6 with the MAML5 adaptation – suggesting that the USL and MAML5 meta-learning algorithms might learn different representations. For USL to have a fair comparison during meta-test time when using the MAML adaptation, we provide the MAML final layer learned initialization parameters to the USL model (but any is fine due to convexity when using a fixed feature extractor). This is needed since during meta-training USL is trained with a union of all the labels (64) – so it does not even have the right output size of 5 for few-shot prediction. Meta-testing was done in the standard 5-way, 5-shot regime.

of MAML and USL to be larger for smaller models and the difference to decrease as the model size increased. Once the two models were, of the same size but large enough, we hypothesized that the meta-test accuracy would be the same. We tested this to rule out that our observations were a consequence of the model size. The results were negative and surprisingly the equivalence between MAML and USL seems to hold even as the model increased – strengthening our hypothesis that the low task diversity might be a bigger factor explaining our observations. We show this in figure 3, and we want to draw attention to the fact this statistical equivalence holds even when using only four filters – the case where we expected the biggest difference.

## 4.4 MAML learns a different base model compared to Union Supervised Learned models – even in the presence of low task diversity

The first four layers of figure 2 shows how large the distance is of a MAML representation compared to a SL representation. In particular, it is much larger than the distance value in the range $[0, 0.1]$ from previous work that compared MAML vs. adapted MAML [18]. We reproduced that and indeed MAML vs. adapted MAML has a small difference (smaller for us) – supporting our observations that a MAML vs. a USL learned representations are different at the feature extractor layer even when the diversity is low. Results are statistically significant.

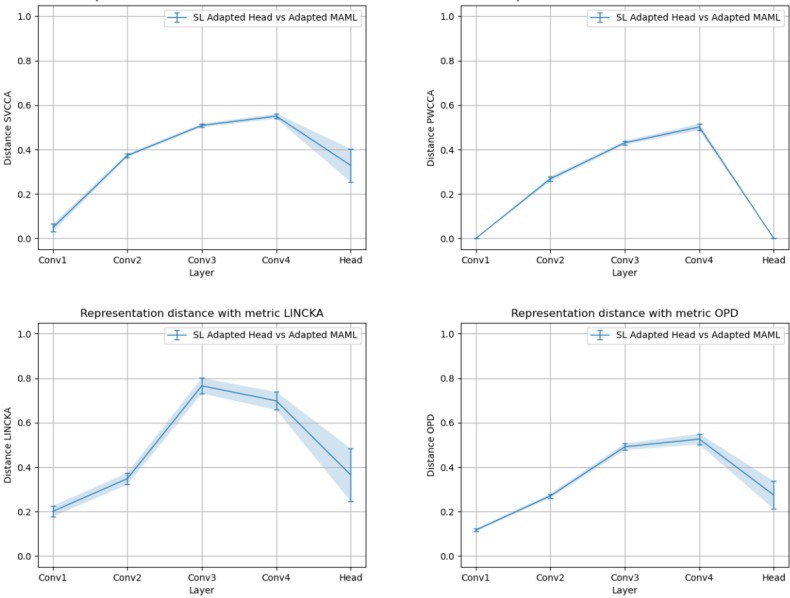

Figure 2: **The classification layer of transfer learning and a MAML5 model decrease in distance – implying similar predictions.** More precisely, an initialization trained with 5 inner steps (MAML5) has an increasingly similar head (classifier) after adaptation with MAML5 compared to the classifier layer of the union supervise learned (USL) model that has been adapted only at the final layer. In particular, the USL model has been adapted with Logistic Regression (LR) with LBFGS with the default value for the l2 regularization parameter given by Python's Sklearn (as in [19]). We showed this trend with four different distance metrics SVCCA, PWCCA, LICKA, and OPD referenced in section 2. Observe that according to PWCCA the distance between the predictions is zero. This is true because the distance of classification layer (indicated as "head" in the figure) is zero. The architecture used here is a five layer CNN as in [15; 25] with their same setup. The benchmark used for this analysis is MiniImagenet.

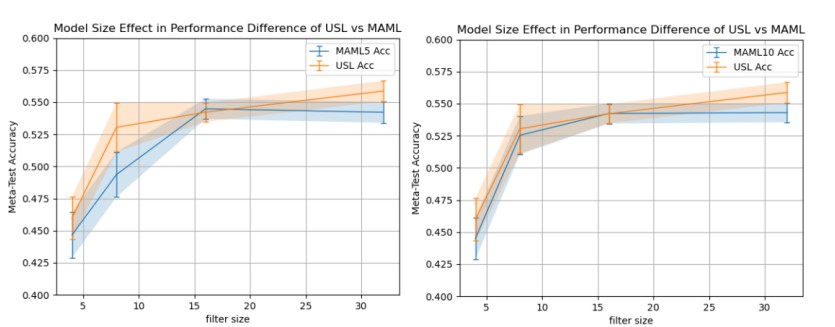

Figure 3: **The meta-test accuracy of MAML and transfer learning using USL is similar in a statistically significant way – regardless of the model size**. In this experiment, we used the MiniImagenet benchmark, the five layer CNN used in [15; 25], and only increased the filter size using sizes 4, 8, 16, and 32. We made sure the comparison was fair by using the same architecture, optimizer, and trained all models to convergence. During meta-training, the MAML model was trained using 5 inner steps. The legends indicating MAMl5 and MAML10 refer to the number of inner steps used at test time. We used a (meta) batch size of 100 tasks.

### 4.5 Synthetic Experiments showing closeness of MAML and Transfer Learning as Diversity Changes

In this section, we show the closeness of MAML and transfer learning (with USL) for synthetic experiments for low and high diversity regimes in Figure 4. In the low regime, the two methods are equivalent in a statistically significant way – which supports the main claims of our paper. As the diversity increases, however, the difference between USL and MAML increases (in favor of USL). This will be explored further in future work.

The task is the usual n-way, k-shot tasks, but the data comes from a Gaussian and the meta-learners are tasked with classifying from which Gaussian the data points came from in a few-shot learning manner. Benchmarks are created by sampling a Gaussian distribution with means moving away from the origin as the benchmark changes. Therefore, the Gaussian benchmark with the highest diversity coefficient has Gaussians that are the furthest from the origin. We computed the diversity coefficient using a proper distance between distributions using the Hellinger distance eluded in section 3 instead of the FIM distance. We can do this because we know the ground truth distribution in our synthetic experiments, and Gaussians have a closed form Hellinger distance. Details on the n-way Gaussian benchmark and diversity coefficient using the Hellinger distance can be found in supplementary section E and F.

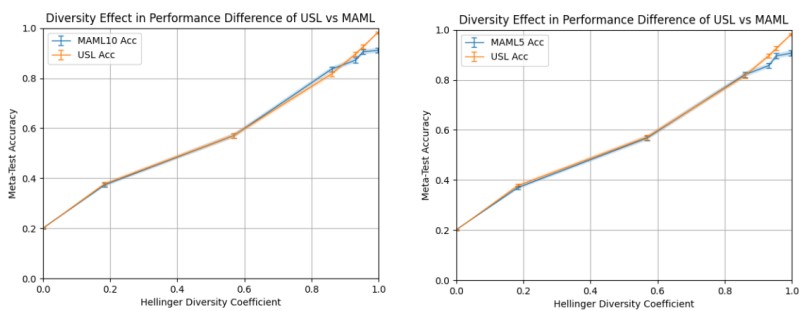

Figure 4: **The meta-test accuracy of MAML and transfer learning using USL is similar in a statistically equivalent way in the low diversity regime in the 5-way, 10-shot Gaussian Benchmarks**. MAML models were trained with 5 inner steps. MAML5 and MAML10 indicate the adaptation procedure at test time. Results used a (meta) batch-size of 500 tasks and 95% confidence intervals. As the diversity of the benchmark increases, the Gaussian tasks are sampled further away from the origin. Note, as the diversity increases, the difference between USL and MAML increases (in favor of USL).

## 5 Related Work

Our work proposes a problem-centric framework for the analysis of meta-learning algorithms inspired from previous puzzling results (19). We propose to use a pair-wise distance between tasks and analyze how this metric might correlate with meta-learning. The closest line of work for this is the long line of work by (20) where they suggest methods to analyze the complexity of a task, propose unsymmetrical distance metrics for data sets, reachability of tasks with SGD, ways to embed entire data sets and more (20; 29; 30; 31). We believe this line of work to be very fruitful and hope that more people adopt tools like the ones they suggest and we propose in this paper before researching or deploying meta-learning algorithms. We hope this helps meta-learning methods succeed in practice – since cognitive science suggests meta-learning is a powerful method humans use to learn (32). In the future, we hope to compare (20)'s distance metrics between tasks with ours to provide a further unified understanding of meta-learning and transfer learning. A contrast between their work and ours is that we focus our analysis from a meta-learning perspective applied to few-shot learning – while their focus is understanding transfer learning methods between data sets.

The use of a distance metric in our definition of the diversity coefficient is inspired by the analysis done by (18). They showed that MAML functions mainly via feature re-use than by rapid learning i.e., that a model trained with MAML changes very little after the MAML adaptation. The main difference

of their work with our is: 1) that we compare MAML trained models against union supervised learned models (USL) instead of only comparing MAML against adapted MAML, and 2) that we explicitly analyzed properties of the data sets. In addition, we use a large set of distance metrics for our analysis including: SVCCA, PWCCA, LINCKA and OPD as proposed by (21; 22; 23; 24).

Our work is most influenced by previous work suggesting modern meta-learning requires rethinking (19). The main difference of our work with theirs is that we analyzed the internal representation of the meta-learning algorithms and contextualize these with quantifiable metrics of the problem being solved. Unlike their work, we focused on a fair comparison between meta-learning methods by ensuring the same neural network backbone was used. Another difference is that they gained further accuracy gains by using distillation – a method we did not analyze and leave for future work.

A related line of work (33; 26) first showed that there exist synthetic data sets that are capable of exhibiting higher degrees of adaptation as compared to the original work by (18). The difference is that they did not compare MAML models against transfer learning methods like we did here. Instead, they focused on comparing adapted MAML models vs. unadapted MAML models.

Another related line of work is the predictability of adversarial transferability and transfer learning. They show this both theoretically and with extensive experiments (34). The main difference between their work and ours is that they focus their analysis mainly on transfer learning, while we concentrated on meta-learning for few-shot learning. In addition, we did not consider adversarial transferability – while that was a central piece of their analysis. Further, related work is outlined in the supplementary section I.

## 6   Discussion and Future Work

In this work, we presented a problem-centric framework when comparing transfer learning methods with meta-learning algorithms – using USL and MAML as the canonical representatives of transfer and meta-learning methods respectively. We showed the diversity coefficient of the popular MiniImagenet and Cifar-fs benchmark is low and that under a fair comparison – MAML is very similar to transfer learning with USL. This was also true even when decreasing the model size – removing the alternative hypothesis that the equivalence of MAML and transfer learning with USL held due to large models. Instead, this suggests strengthens our hypothesis that the diversity of the data might be the driving factor. The equivalence of MAML and USL also replicated in our synthetic experiments. Therefore, we challenge the suggestions from previous work (19) that only a good embedding can beat more effective than sophisticated meta-learning – especially in the low diversity regime. In addition, our synthetic experiments show a promising scenario where we can systematically differentiate meta-learning algorithms from transfer learning algorithms – which supports our actionable suggestion to use the diversity coefficient to effectively study meta-learning and transfer learning algorithms. We hope to study this in more depth in the future with real and synthetic data.

We also have theoretical results from a statistical decision perspective in the supplementary section **??** that inspired this work and suggest that when the distance between tasks is zero – then the predictions of transfer learning, meta-learning and even a fixed model with no adaptation are all equivalent (with the l2 loss). The results are theoretically limited because we can only reason when the diversity is exactly zero, but regardless provided an interesting perspective to study and inspire empirical work.

We hope this work inspires the community in meta-learning and machine learning to construct benchmarks from a problem-centric perspective – that go beyond large scale data sets – using have quantitative metrics.

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
