# A  Further Discussions

We'd like to emphasize that our synthetic experiments are promising because we can systematically differentiate meta-learning algorithms from transfer learning algorithms – which supports our actionable suggestion to: 1. use the diversity coefficient to effectively study meta-learning and transfer learning algorithms, and 2. to use the diversity coefficient to design better benchmarks. In addition, this problematizes the observations that fo-proto-MAML in meta-data set [44] is better than transfer learning solutions – since our synthetic experiments show MAML is not better than USL in the high diversity regime. To further problematize, we want to point out that meta-learning methods are not better than transfer learning as observed by [20] – as observed in our synthetic experiments. We hypothesize however that the two scenarios in [44] are different [20]. The first one focuses on the same meta-training and meta-testing conditions, while the latter focuses on a cross-domain. We hypothesize that the cross-domain scenario might benefit from a meta-learning which lower variance (e.g., a fixed embedding [43]) – which might explain why sophisticated meta-learning solutions might perform worse on the cross-domain setting as observed in [20]. Further research is needed in both benchmarks – especially from a problem centric perspective with quantitative methods like the ones we suggest.

In addition, we hypothesize that diversity might be a good proxy to predict the difference between meta-learning and transfer learning methods. More precisely, we conjecture that in a low diversity setting meta-learning methods are equivalent at meta-test time to transfer learning methods but their difference increases as the diversity of tasks in a benchmark increases. In the high diversity regime we conjecture that the difference between meta-learning and transfer learning methods increases as the diversity increases. We are optimistic that meta-learning algorithms might outperform transfer learning methods, once we start comparing them in more thoughtfully designed benchmarks. It is possible that despite our efforts, meta-learning algorithms – as currently designed – are too sophisticated and in fact lead to meta-overfitting, as shown in previous work [32].

We'd like to emphasize, that up until now, the meta-learning community has evaluated meta-learning algorithms in benchmarks that might not be the most appropriate. We conjecture high diversity benchmarks are more appropriate, since they might capture the meta-learning inductive prior: high diversity means that adaptation is required by construction. Thus, we conjecture that previous conclusions should be taken with a grain of salt until a more in depth study can be made in the high diversity regime – especially with benchmarks with real world data that have been analyzed extensively with metrics like the diversity coefficient that we propose. We conjecture that we can finally do meta-learning research effectively – given that a regime where meta-learning and transfer learning methods can be differentiated has been discovered and previous low diversity benchmarks have been understood.

We also conjecture that meta-learning research is different from classical machine learning research. Historically, a seminal paper is the one where AlexNet was proposed [25]. In that time we had low performance on a fixed task e.g., Imagenet and couldn't even interpolate the data (i.e., reach zero train error). We conjecture that meta-learning is different because if we have a diversity so large where all possible tasks are incorporated, then we should reach the no-free lunch theorem regime [46] – where all algorithms should perform the same on average. Therefore, we hypothesize that a deliberate and quantitative efforts to design benchmarks is essential. A great example of such an attempt is the Abstraction and Reasoning Corpus (ARC) benchmark [11] – which was made very thoughtfully with Artificial General Intelligence (AGI) in mind. We conjecture meta-learning is the most promising path in that direction, and hope this work inspires the design of benchmarks that lead to actionable and deliberate attempts to make progress to build such AGI technologies.

# B  Related Work (continued)

The meta-learning literature is growing quickly and hope to provide a wider coverage here.

In terms of benchmarks, we'd like to start with the ARC benchmark [11]. ARC was designed with AGI in mind – arguably the ultimate meta-learner. Its focus is primarily on visual reasoning using program synthesis techniques. We hypothesize it's a very promising path but our work inspires extension that go beyond program synthesis approaches. The meta-data set benchmark is an attempt to make the data set for few-shot learning at a larger scale and more diverse [44]. The main difference

of their work and ours is that we propose a quantitative metric to measure the intrinsic diversity in the data and go beyond data set size or number of classes. They also showed that a meta-learning algorithms – fo-Proto-MAML – is capable of beating transfer learning. However, they also showed transfer learning baselines are in fact quite difficult to beat. The IBM Cross-Domain few-shot learning benchmark [20] is a fascinating benchmark to evaluate meta-learning algorithms. Their central premise however is transfer from a source domain to a different target domain – instead of our setting where tasks are created from the same meta-distribution. This is why their paper is considered, in addition to few-shot learning, a *cross-domain benchmark*. We believe this is an essential scenario to think about, but consider it different from our setting or the setting of meta-data set. We'd also like to emphasize that they do not employ a metric like our diversity coefficient that quantitatively asses the diversity of their benchmarks. These two last benchmarks, although fascinating, are missing the essential quantitative analysis of the data itself we are trying to propose.

The work by [8] give to the best of our knowledge – the first non-vacuous generalization bounds for the (supervised) meta-learning setting. Their statements apply to non-convex loss function and use stability theory at the task level. The bound depends on the mutual information on the input data vs the output data of the meta-learner. The results, although fascinating, are not built to separate classes of meta-learning – like our work attempts to do empirically.

The work by [45] proposes the idea of global labels as a way to indirectly optimize for the met—learning objective for a fixed feature extractor. Global labels is equivalent to the concept we call USL in this paper. They show that pre-training (i.e. using USL/global labels) provides excellent meta-test results – including with their method (named MeLa) that can infer global labels given only local labels provided at in episodic meta-training. Their theoretical analysis depends on a fixed feature extractor, instead of considering the whole end-to-end meta-learner as a whole – meaning two different deep learning models cannot be used in their analysis. Therefore, their analysis fails to separate how different feature extractors might be trained, e.g. comparing USL vs MAML directly in an end-to-end fashion. In contrast, we instead tackle this question head on theoretically L (with limited results) but instead show that the feature extractors indeed are different empirically E.

The work by [13] proposes a theoretical treatment of meta-learning using meta-learners with closed-form equations derived from ridged regularization using fixed features. They formulate the conditional and unconditional formulation using side information for the task (e.g. the support set) and show the conditional method is superior. In relation to our work, they do not provide characterizations of the role of a neural network doing end-to-end meta-learning (in their empirical or theoretical analysis). In contrast, our findings make an explicit effort in understanding the role of the neural network in meta-learning in an end-to-end fashion through emperical analysis. Another contrast is that their results are highly theoretical, while ours focus on empirical results. In addition, their results are on synthetic experiments and do not explore their findings in the context of modern few-shot learning benchmarks like MiniImagenet or Cifar-fs.

The work by [19] provide strong evidence that adaptation at test time is best done when the meta-trained model matches the adaptation it was meta-trained with. This is shown because their classically pre-trained nets cannot perform better than the MetaOpt models with any fine-tuning method. However, their results cannot beat [43] and thus does not help separate the role of meta-training and union supervised learning (USL). Their Resnet12 results do provide further support to our hypothesis that large enough neural networks all perform the same, since 78.63 (Goldblum) vs 79.74 (RFS) have very close errors, in line with our findings. However, we hypothesize it is not due to the model size in accordance with our experiments 3.

The work by [18] provides theoretical bounds of when the expected risk of MAML and DRS (Domain Randomized Search) by bounding the gradient norm. DRS attempts to model USL but fails to do so completely, because USL is capable of modeling adaptation because the final layer is capable of adaption. Thus, it does not address the capabilities of the feature extractor being able to learn all the information needed to meta-learn. Concisely, their analysis is not capable of separating performance of MAML and USL. Even if it hypothetically could, their analysis remains an upper bounds (with assumptions). This raising the question if their method truly explain the observations that transfer learning methods – like USL – beat meta-learning methods. In addition, they do not provide in depth empirical analysis with respect to any real few-shot learning benchmarks like MiniImagent or Cirfar-fs.

The work by [26] provides an exploration of the effects of diversity in meta-learning. The main difference with our work is that they focus mostly on sampling strategies, and it's effect on diversity, while we focused on the intrinsic diversity in the benchmarks themselves.

The work by [40] provides a theoretical analysis on the difference between interpolation and extrapolation in transfer learning (and domain generalization). We believe this type of theory may be helpful as an inspiration to explore why in the high diversity regime there seems to be a difference between the performance of meta-learning and transfer learning methods.

## C  Convergence of Learning Curves for Fair Comparison

### C.1  Convergence of Learning Curves for MiniImagenet and Cifar-fs

In this section, we have the plots showing the learning curves achieving convergence for the models used in figure 1 and 2. Note, the learning curves for the models trained with MAML look noisier because the distributed training reduces the size (meta) batch size for logging purposes. In addition, due to episodic meta-training, (meta) batch sizes have to be smaller compared to batch sizes used in USL.

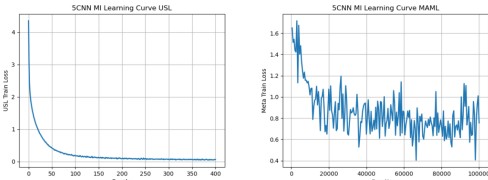

Figure 5:  Plot showing convergence of 5CNN on MiniImagenet.

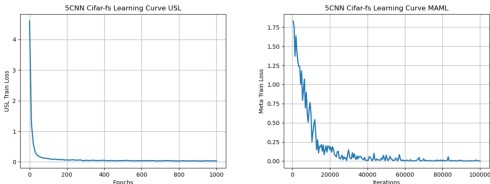

Figure 6:  Plot showing convergence of 5CNN on Cifar-fs.

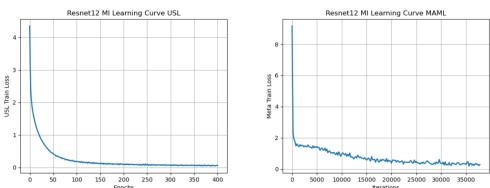

Figure 7:  Plot showing convergence of Resnet12 on MiniImagenet.

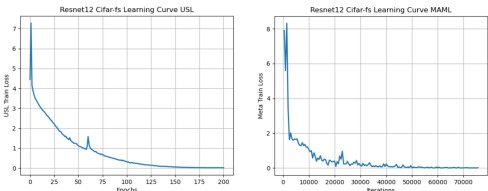

Figure 8:  Plot showing convergence of Resnet12 on Cifar-fs.

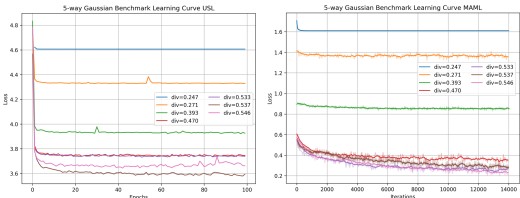

Figure 9: Plot showing convergence of a custom 3-layer fully connected network, for all synthetic Gaussian benchmarks tested. Each curve represents a different synthetic Gaussian benchmark tested on either MAML (left plot) or USL (right plot). The curves are then color-coded by the value of the Task2Vec-based task diversity coefficient of the Gaussian benchmark tested in that particular run.

## D Experimental Details

### D.1 Experimental Details on MiniImagenet and Cifar-fs

We will explain the details for the four models we trained on figure 1 and 2.

**Summary:** We trained a five layer CNN (5CNN) and Resnet12 on both MiniImagenet and Cifar-fs to convergence. We used the Adam optimizer with learning rate 1e-3. We used the standard MiniImagenet and Cifar-fs data augmentations as provided in [5] matching [43].

**Experimental Details for 5CNN on MiniImagent:** We used the five layer CNN from [17, 39]. We used 32 filters as used in previous work. We used the Adam optimizer with learning rate 1e-3 for both MAML and USL. We used no scheduler. We trained the USL model for 1000 epochs. We trained the MAML model for 100,000 episodic iterations (outer loop iterations). We used a batch size of 128 for USL and a (meta) batch size of 8 for MAML. For MAML we used an inner learning rate of 1e-1 and 5 inner learning steps. We did not use first order MAML. It took 3 hours 5 minutes 5 seconds to train USL to convergence with a single GPU. It took 1 day 6 hours 21 minutes 8 seconds to train MAML to convergence with 4 NVIDIA GeForce GTX TITAN X GPUs.

**Experimental Details for Resnet12 for MiniImagent:** We used the Resnet12 provided by [43]. We used the Adam optimizer with learning rate 1e-3 for both MAML and USL. We used the same cosine scheduler as in [43] for USL and no cosine scheduler for MAML. We trained the USL model for 186 epochs. We trained the MAML model for 37,800 episodic iterations (outer loop iterations). We used a batch size of 512 for USL and a (meta) batch size of 4 for MAML. For MAML we used an inner learning rate of 1e-1 and 4 inner learning steps. We did not use first order MAML. It took 1 day 17 hours 2 minutes 41 seconds to train USL to convergence with a single dgx A100-SXM4-40GB GPU. The MAML model was trained with Torchmeta [12] which didn't support multi gpu training when we ran this experiment, so we estimate it took 1-2 weeks to train on a single GPU. In addition, it was ran with an earlier version of our code, so we unfortunately did not record the type of GPU but suspect it was either an A100, A40 or Quadro RTX 6000.

**Experimental Details for 5CNN for Cifar-fs:** We used the five layer CNN from [17, 39] provided by [5]. But we used 1024 filters instead of 32 (to speed up convergence). We used the Adam optimizer with learning rate 1e-3 for both MAML and USL. We used the same cosine scheduler as in [43] for MAML and no cosine scheduler for USL. We trained the USL model for 1000 epochs. We trained the MAML model for 100,000 episodic iterations (outer loop iterations). We used a batch size of 256 for USL and a (meta) batch size of 8 for MAML. For MAML we used an inner learning rate of 1e-1 and 5 inner learning steps. We did not use first order MAML. It took 10 hours 43 minutes 31 seconds to train USL to convergence with a single GPU dgx A100-SXM4-40GB. It took 2 days 9 hours 26 minutes 27 seconds to train MAML to convergence with 4 Quadro RTX 6000 GPUs.

**Experimental Details for Resnet12 for Cifar-fs:** We used the Resnet12 provided by [43]. We used the Adam optimizer with learning rate 1e-3 for both MAML and USL. We used the cosine scheduler used in [43] for both USL and MAML. We trained the USL model for 200 epochs. We trained the MAML model for 75,500 episodic iterations (outer loop iterations). We used a batch size of 1024 for USL and a (meta) batch size of 8 for MAML. For MAML we used an inner learning rate of 1e-1 and 5 inner learning steps. We did not use first order MAML. It took 45 minutes 54 seconds to train USL

to convergence with a single GPU. It took 1 day 19 hours 29 minutes 31 seconds to train MAML to convergence with 4 dgx A100-SXM4-40GB GPUs.

**Why the Adam optimizer?** We hypothesize that the Adam optimizer is the most appropriate optimizer for a fair comparison for various reasons. First, the Adam optimizer is widely used – making our results most relevant and broadly applicable. Adam is generally a stable optimizer – especially for sophisticated meta-learning algorithms like MAML. It is not uncommon to have SGD result in exploding gradients or end up diverging – especially for MAML. Most importantly however, we hope to stay faithful whenever possible to how modern transformer models are trained – because they have been shown to be good meta-learners, e.g., gpt-3 is often cited as a zero-shot learner [7]. These type of models do use more complicated learning schemes besides only Adam (e.g., warm-ups, decay rates etc.) but we hypothesize using Adam is a good first step. We conjecture that the small benefits that SGD might provide are negligible compared to the stability that Adam provides, especially as the scale of the data sets starts to increase. Without Adam we conjecture it would be hard to even perfectly fit the data for large scale data sets as it's usually done in Deep Learning. This was definitively true in our own experiments. Therefore, we decided to use Adam for our experiments, since it would be too hard to use SGD reliably at scale or with sophisticated meta-learning algorithms.

## D.2 Experimental Details on N-way Gaussian Tasks

We used a custom 3-layer fully connected network derived from Learn2Learn's OmniglotFC model [5], with parameters $input\_size = 1$, $output\_size = 5$, and hidden layer sizes $sizes = [128, 128]$. We used the Adam optimizer with learning rate 1e-3 for both MAML and USL, but did not use a cosine scheduler for either USL or MAML. We trained the USL model for 100 epochs. We trained the MAML model for 14,000 episodic iterations (outer loop iterations). We used a batch size of 100 for USL and a (meta) batch size of 100 for MAML. For MAML we used an inner learning rate of 1e-1 and 5 inner learning steps. We did not use first order MAML. It took 19 minutes 24 seconds to train USL to convergence with a single Titan X. It took 2 days and 13 hours 6 minutes 27 seconds to train MAML to convergence with a single Titan X.

# E Feature Extractor Analysis of USL and MAML

Figure 10 significant difference between the feature extractor layers of a MAML trained model vs. a union supervised learned model.

# F Background of Few-shot Learning Basics

The goal of few-shot learning is to learn to classify from a limited set of training samples. A few-shot benchmark is utilized to evaluate few-shot learning algorithms and typically contains many classes and a smaller number of samples per class. Typically, few-shot learning algorithms learn in episodes, where in each episode, a task consisting of a train (or support) set and a held-out validation (or query) set is sampled. In particular, a task is a $n$-way $k$-shot classification problem, means that the support and query sets each consist of $n$ classes sampled from the benchmark, and each of the $n$ classes are represented by $k$ shots or examples. The learner uses the support set to adapt to the task, and the query set to evaluate the performance on the given task.

# G Synthetic Gaussian Benchmark and N-way Gaussian Tasks

We create a series of synthetic few-shot benchmarks, where each Gaussian benchmark $B$ is defined by four parameters $B = (\mu_m, \sigma_m, \mu_s, \sigma_s)$. To form the dataset of our benchmark, we first sample 100 meta-train, 100 meta-test, and 100 meta-validation classes, where class $1 \leq i \leq 300$ is a Gaussian parameterized by $(\mu_{class_i}, \sigma_{class_i})$ where

$$\mu_{class_i} \sim N(\mu_m, \sigma_m), \sigma_{class_i} \sim |N(\mu_s, \sigma_s)|$$

Then, for each class $i$, we sample 1000 data points $(x_{i,1}, i) \ldots (x_{i,1000}, i)$ where each datapoint $(x, y)$ is composed of a input value $x \in \mathbb{R}$ and class label $1 \leq y \leq 300$. The input values $x_{i,1} \ldots x_{1,1000}$

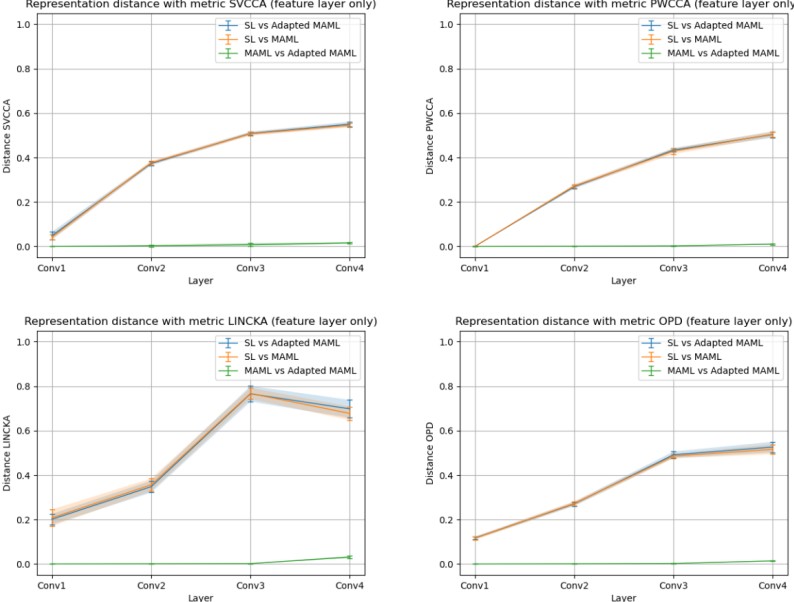

Figure 10: **Shows the significant difference between the feature extractor layers of a MAML trained model vs. a union supervised learned model – especially in contrast to the small change in the adapted MAML model (green line).** This figure suggests that although benchmark diversity is small, a meta-learned representation still learns through a different mechanism than a supervised learned representation. Note that the green line is our reproduction of previous work [37] that showed that a MAML trained model does not change after using the MAML adaptation. They term this observation as "feature re-use".

are each sampled from class $i$'s class distribution:

$$x_{i,1} \dots x_{i,1000} \sim N(\mu_{class_i}, \sigma_{class_i})$$

Having defined our dataset underlying our benchmark, we may now sample individual tasks from our benchmark. Each task in our benchmark is 5-way, 10-shot - that is, each task is formed by first sampling 5 ways from the benchmark dataset, then sampling 10 shots from each of the 5 ways. The goal of each task is to correctly predict which of the 5 ways an input value $x \in \mathbb{R}$ falls into. We conducted experiments using 7 different benchmarks, with each benchmark defined by four parameters and its corresponding Hellinger distribution diversity coefficient and Task2Vec task diversity coefficient, as listed in Table 3:

## H    Distribution-based Diversity Metrics

In addition to the task diversity methods (such as Task2Vec) that we chose as a measure of diversity across our experiments in our main paper, we would like to introduce an additional class of diversity metrics that we call *distribution diversity*. Unlike task diversity, which quantifies diversity through the expected distance between any two distinct tasks sampled from the benchmark, distribution diversity quantifies diversity through the expected distance between any two distinct *distributions* that underlie the benchmark. In our synthetic Gaussian experiments, we define the distribution diversity of our Gaussian benchmark as the expected Hellinger distance between two distinct Gaussian class distributions sampled from the benchmark - we describe the calculation of the distribution diversity of our synthetic Gaussian benchmark in more detail in Section I.

## I    Hellinger Diversity Coefficient and Hellinger Distance

An alternative metric to the Task2Vec-based task diversity metric is the Hellinger-based distribution diversity metric. The Hellinger-based distribution diversity of our Gaussian benchmark is obtained

| Benchmark Parameters $(\mu_m, \sigma_m, \mu_s, \sigma_s)$ | Hellinger-based Distribution Diversity | Task2Vec-based Task Diversity |
|---|---|---|
| (0, 0.01, 1, 0.01) | 7.475e-05 $\pm$ 4.891e-07 | 0.247 $\pm$ 1.04e-3 |
| (0, 1, 1, 0.01) | 0.183 $\pm$ 1.24e-3 | 0.271 $\pm$ 1.15e-3 |
| (0, 3, 1, 0.01) | 0.574 $\pm$ 2.28e-3 | 0.393 $\pm$ 1.79e-3 |
| (0, 10, 1, 0.01) | 0.860 $\pm$ 1.75e-3 | 0.470 $\pm$ 2.35e-3 |
| (0, 20, 1, 0.01) | 0.929 $\pm$ 1.31e-3 | 0.533 $\pm$ 2.47e-3 |
| (0, 30, 1, 0.01) | 0.952 $\pm$ 1.10e-3 | 0.537 $\pm$ 2.57e-3 |
| (0, 1000, 1, 0.01) | 0.998 $\pm$ 2.07e-4 | 0.546 $\pm$ 2.74e-3 |

Table 3: **Benchmarks of increasing diversity are created by increasing $\sigma_m$, or the standard deviation of the class mean** A larger $\sigma_m$ increases the variance of the class means, making their respective class distributions farther apart on average and causing both the Hellinger-based distribution diversity and Task2Vec-based task diversity coefficients to increase. We varied $\sigma_m$ from 0.01 to 1000 and fixed all remaining benchmark parameters to obtain 7 different Gaussian benchmarks. The corresponding Hellinger-based distribution diversity coefficients were obtained by numerically approximating the expected Hellinger distance between two classes sampled from the benchmark and computing the 95% confidence interval of the approximation. We also computed Task2Vec-based task diversity coefficients as an alternative measure to diversity using a random 3-layer fully connected probe network described in Section D. Figure 12 visualizes the Task2Vec task diversities among the synthetic benchmarks via a heatmap showing the relative pairwise distance between sampled tasks.

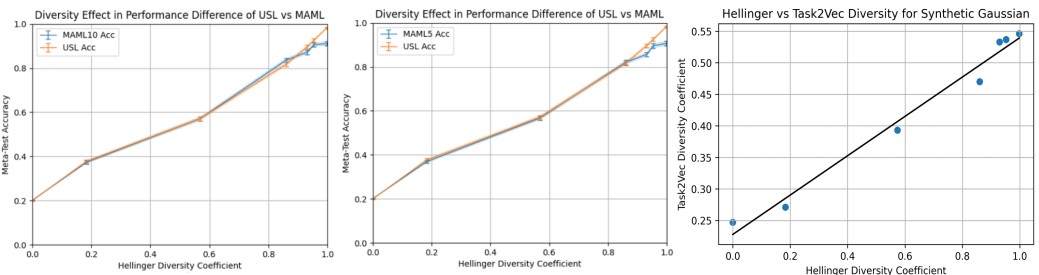

Figure 11: **Shows the strong relation between Hellinger distribution diversity and the Task2Vec task diversity coefficients, as both coefficients may be used interchangeably as a measure for the diversity of a given synthetic Gaussian benchmark** The first two plots show the relation between the Hellinger-based distribution diversity of a synthetic Gaussian benchmark and the benchmark's performance on the MAML5, MAML10, and USL methods. These first two plots are noticeably similar to Figure 4 (where Task2Vec-based task diversity was used as a measure of diversity instead of Hellinger-based distribution diversity), which indicates that our Hellinger-based distribution diversity metric also serves as a good proxy for task diversity. The rightmost plot shows a strong positive correlation between Hellinger-based distribution diversity and Task2Vec task diversity (Pearson $r = 0.990$).

by computing the expected Hellinger distance between any two classes sampled from the benchmark. That is, for some benchmark parameterized by $B = (\mu_m, \sigma_m, \mu_s, \sigma_s)$, the diversity coefficient is given by

$$div(B) = \mathbb{E}_{\mu_1, \mu_2 \sim N(\mu_m, \sigma_m)} \mathbb{E}_{\sigma_1, \sigma_2 \sim |N(\mu_s, \sigma_s)|} [H^2(N(\mu_1, \sigma_1), N(\mu_2, \sigma_2))]$$

739 where $H^2$ denotes the squared Hellinger distance metric and $N(\mu_1, \sigma_1), N(\mu_2, \sigma_2)$ denote the
740 distributions of the two classes sampled from the benchmark. The Hellinger-based distribution
741 diversity metric provides an intuitive, model-agnostic characterization of the diversity of a benchmark
742 - the larger the diversity, the less similar any two classes within the benchmark are, and the easier it is
743 to distinguish between two classes. Conversely, the lower the diversity, the more similar any two
744 classes within the benchmark are, and the harder it is to distinguish between two classes due to a
745 larger overlap between the two classes' distributions.
746

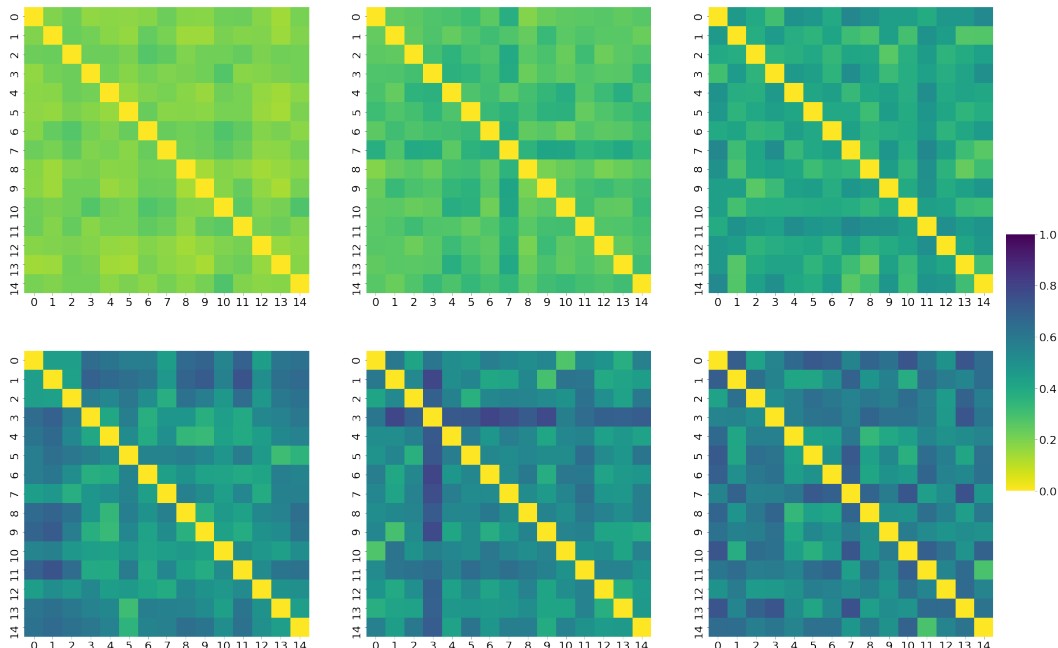

Figure 12: **Heatmaps show how benchmarks with larger Task2Vec-based task diversity coefficient show more heterogeneity between sampled tasks** Each heatmap below shows the pairwise distance between fifteen 5-way, 10-shot few-shot learning tasks sampled from the various synthetic Gaussian benchmarks described in Table 3. Note that as $\sigma_m$ (the standard deviation of the class mean) increases, the distance between two tasks becomes larger on average and more varied, which can be seen as the heatmaps become more heterogeneous. This increase in expected distance among different tasks in turn increases the Task2Vec-based task diversity coefficient, which summarizes the average distance between any two tasks. From left to right, top to bottom, the benchmarks tested have parameters $\sigma_m = 0.01, 1, 3, 20, 30, 1000$ and Task2Vec-based task diversity coefficient parameters $div = 0.247, 0.271, 0.393, 0.533, 0.537, 0.546$.

Note that the closed-form equation for the Hellinger distance between the two class distributions $N(\mu_1, \sigma_1), N(\mu_2, \sigma_2)$ is given by

$$H^2(N(\mu_1, \sigma_1), N(\mu_2, \sigma_2)) = 1 - \sqrt{\frac{2\sigma_1\sigma_2}{\sigma_1^2 + \sigma_2^2}} e^{-\frac{1}{4}\frac{(\mu_1-\mu_2)^2}{\sigma_1^2+\sigma_2^2}}$$

However, there is no simple closed-form equation for computing the diversity $div(B)$ itself. As a result, we computed the diversity coefficient as a numerical approximation by repeatedly sampling two classes from the benchmark distribution and calculating the Hellinger distance between the two classes. These samples ultimately provide a 95% confidence interval that represents the expected Hellinger distance between two classes sampled from the benchmark.

We also compared our Hellinger-based distribution diversity coefficient with the Task2Vec-based task diversity coefficient for each of the synthetic Gaussian benchmarks tested in Table 3. We observe a strong positive correlation between the Hellinger-based distribution diversity and Task2Vec-based task diversity coefficients according to Figure 11, indicating that the Hellinger-based distribution diversity serves as a effective proxy for task diversity when the number of ways and shots of all tasks are fixed.

## J Analysis of distribution of task distances in few-shot learning benchmarks

### J.1 Heat Maps show Low Diversity and Homogeneity of tasks from MiniImagenet and Cifar-fs

In this section, we show the heat maps showing the distances between 5-way, 20-shot few-shot learning tasks from MiniImagenet and Cifar-fs in figure 13 and 14. We used 20-shots because we do not need to separate the data into support and query set to compute the diversity coefficient. We show that tasks sampled from these benchmarks create not only a low diversity coefficient on average, but also at the level of individual distances between pairs of tasks. In addition, the heat map's uniform coloring reveals that it is also justifiable to call the tasks from these benchmarks *homogeneous*. Low diversity is shown because the distances are between 0.07-0.12 given that max is 1.0 and minimum is 0.0.

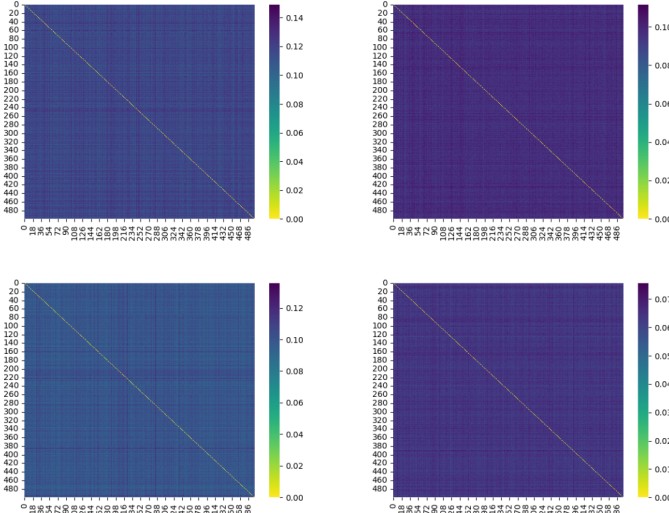

Figure 13: **Shows homogeneity and low diversity of 5-way, 20-shot tasks from MiniImagenet using the Task2Vec distance [4].** The top left heat map uses a Resnet18 pre-trained on Imagenet to compute the Task2Vec distance between tasks. The top right heat map uses a Resnet18 with random weights to compute the Task2Vec distance between tasks. The bottom left heat map uses a Resnet34 pre-trained on Imagenet to compute the Task2Vec distance between tasks. The bottom right heat map uses a Resnet34 with random weights on Imagenet to compute the Task2Vec distance between tasks. Homogeneity is shown because of the uniform color shown in the heat map. Low diversity is shown because the distance is between 0.07-0.12 given that max is 1.0 and minimum is 0.0. Note the diagonal is exactly zero because it is comparing the same tasks. The axis indices indicate the arbitrary name for the tasks. Indices between heat maps do not indicate the same task. We used the cosine distance between task Task2Vec embeddings.

### J.2 Histograms of distances of tasks in the synthetic Gaussian Benchmark, MiniImagenet and Cifar-fs

In this section, we show the histograms of the cosine distances between pairs of tasks for the Gaussian Benchmark, MiniImagenet and Cifar-fs. The main purpose of this is to argue that a (relatively small) sample of the tasks is sufficient to estimate population statistics – like the expected distance between tasks i.e. diversity coefficient.

For ease exposition of the argument, consider the case where we have 500 distance from a large population of size $\binom{64}{5} = 7,624,512$. The goal is to argue that 500 samples are enough to make strong statistical inferences about the population – even if it's as large as $7,624,512$. If we assume the distribution of the data is Gaussian, then we expect to see a single mode with an approximate bell curve. Therefore, if we plot the histogram of task pair distances of the 500 tasks and see this

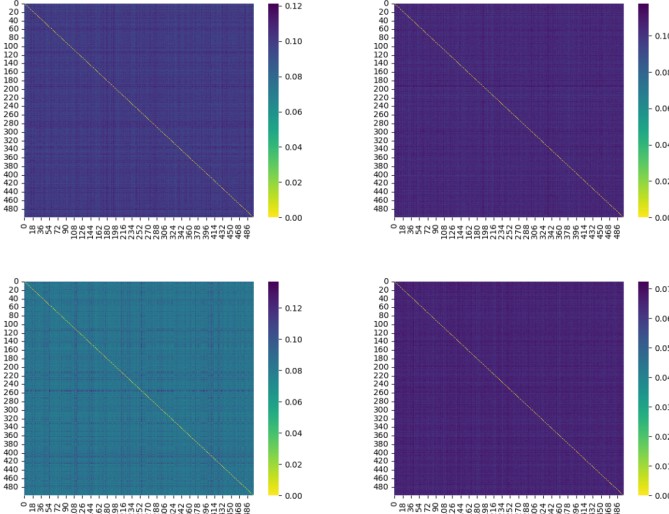

Figure 14: **Shows homogeneity and low diversity of 5-way, 20-shot tasks from Cifar-fs using the Task2Vec distance.** The top left heat map uses a Resnet18 pre-trained on Imagenet to compute the Task2Vec distance between tasks. The top right heat map uses a Resnet18 with random weights to compute the Task2Vec distance between tasks. The bottom left heat map uses a Resnet34 pre-trained on Imagenet to compute the Task2Vec distance between tasks. The bottom right heat map uses a Resnet34 with random weights on Imagenet to compute the Task2Vec distance between tasks. Homogeneity is shown because of the uniform color shown in the heat map. Low diversity is shown because the distance is between 0.07-0.12 given that max is 1.0 and minimum is 0.0. The axis indices indicate the arbitrary name for the tasks. Indices between heat maps do not indicate the same task. We used the cosine distance between task Task2Vec embeddings.

then we can infer our Gaussian assumption is approximately correct. Given that we do see that in figures 15, 16, 17 then we can infer our assumption is approximately correct. This implies we can make strong statistical assumptions about the population – in particular, that we have a good estimate of the diversity coefficient using 500 samples. Additionally, in histograms also discard the presence of outlier tasks.

As a remark, we want to emphasize that if we use $T$ tasks to estimate the diversity coefficient, we in fact will use $\frac{T^2-T}{2} = O(T^2)$ distances to compute the diversity coefficient. This increases the computation cost, but does make the number of sample to compute the mean larger. For a subsample size of, $T = 500$ we in fact use $\frac{T^2-T}{2} = 124,750$ distances to compute the diversity coefficient.

# K    Background on distance metrics

## K.1    Neuron Vectors

The representation of a neuron $d$ in layer $l$ is the vector $z_d^{(l)}(X) \in \mathbb{R}^N$ of activations for a set of $N$ examples, where $X \in \mathbb{R}^{N,D}$ is the data matrix with $N$ examples.

## K.2    Layer Matrix

A layer matrix $L$ for layer $l$ is a matrix of neuron vectors $z_d^{(l)}(X) \in \mathbb{R}^N$ with shape, $[N, D_i]$ i.e. $L \in \mathbb{R}^{N,D_i}$. In other words, the layer matrix $L$ is the subspace of $\mathbb{R}^N$ spanned by its neuron vectors $z_d^{(l)}(X)$. In short, $L$ is the layer matrix $[z_d^l; \ldots; z_{D_1}^l] \in \mathbb{R}^{N,D_i}$ with neuron vector $z_d^l$.

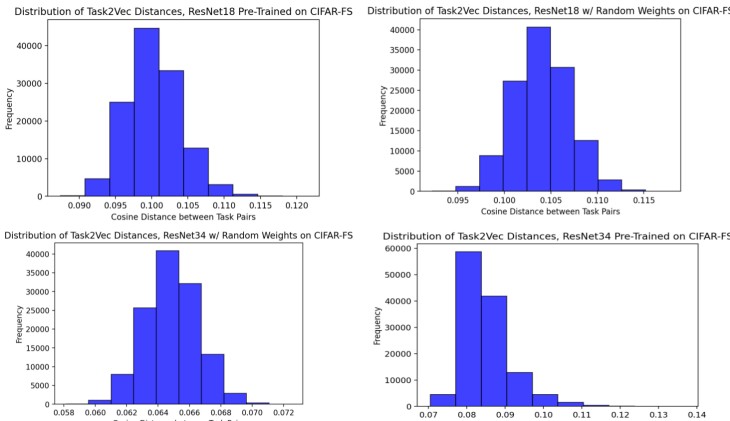

Figure 15: **Histogram of distances of 5-way, 20-shot tasks from Cifar-fs using the Task2Vec distance. This plot justifies the use of a subsample of the population to estimate the diversity coefficient because of its approximate Gaussian distribution.** For the full argument, see the main text, section J.2. The top left histogram uses a Resnet18 pre-trained on Imagenet to compute the Task2Vec distance between tasks. The top right histogram uses a Resnet18 with random weights to compute the Task2Vec distance between tasks. The bottom left histogram uses a Resnet34 pre-trained on Imagenet to compute the Task2Vec distance between tasks. The bottom right histogram uses a Resnet34 with random weights on Imagenet to compute the Task2Vec distance between tasks.

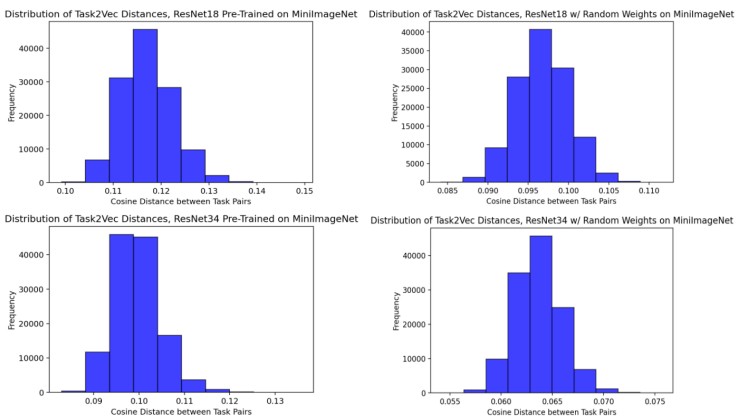

Figure 16: **Histogram of distances of 5-way, 20-shot tasks from MiniImagenet using the Task2Vec distance. This plot justifies the use of a subsample of the population to estimate the diversity coefficient because of its approximate Gaussian distribution.** For the full argument, see the main text, section J.2. The top left histogram uses a Resnet18 pre-trained on Imagenet to compute the Task2Vec distance between tasks. The top right histogram uses a Resnet18 with random weights to compute the Task2Vec distance between tasks. The bottom left histogram uses a Resnet34 pre-trained on Imagenet to compute the Task2Vec distance between tasks. The bottom right histogram uses a Resnet34 with random weights on Imagenet to compute the Task2Vec distance between tasks.

### K.3   CCA

Canonical Correlation Analysis (CCA) is a well established statistical technique for comparing the (linear) correlation of two sets of random variables (or vectors of random variables). In the empirical case, however, one computes the correlations between two sets of data sets (e.g. two matrices $X \in \mathbb{R}^{N,D_1}$ and $Y \in \mathbb{R}^{N,D_2}$ with $N$ examples and $D_1, D_2$ features or layer matrices).

**True distribution based Canonical Correlation Analysis (CCA):** What we call true distribution based CCA is the standard CCA measure using the true but known distribution of the data $p^*(x)$ and $p^*(y)$. In this case, CCA searches for a pair of linear combinations $a^*, b^*$ of two set of random

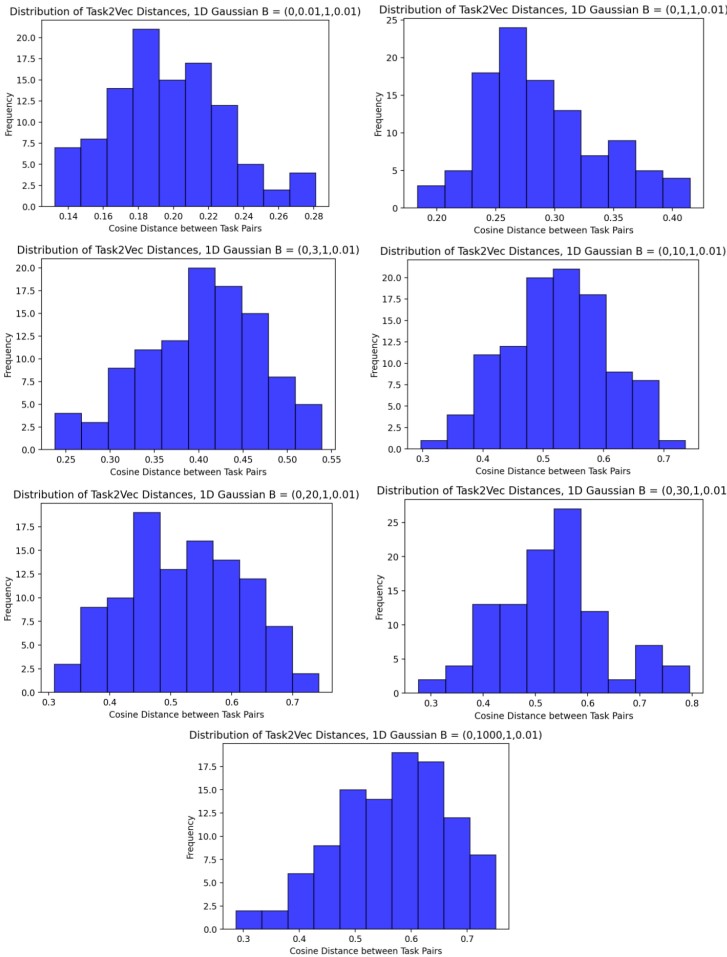

Figure 17: **Histogram of distances of 5-way, 20-shot tasks from the Synthetic Gaussian benchmark using the Task2Vec distance. This plot justifies the use of a subsample of the population to estimate the diversity coefficient because of its approximate Gaussian distribution.** For the full argument, see the main text, section J.2. The meta parameters generating tasks for each benchmark are denoted by $B = (0, x, 1, 0.01)$ where $x$ is in the list $[0.01, 1, 3, 10, 20, 30, 1000]$ indicating the mean to generate the mean of the Gaussian tasks. For full details of the synthetic Gaussian benchmark, see section I.

variables (or vectors of random variables) $\boldsymbol{x} = [X_1, \ldots, X_{D_1}]$ and $\boldsymbol{y} = [Y_1, \ldots, Y_{D_2}]$ that maximizes the Pearson correlation coefficient:

$$a^*, b^* = arg\max_{a,b} \frac{\mathbb{E}_{X,Y}[(a^\top \boldsymbol{x})((b^\top \boldsymbol{y}))]}{\sqrt{\mathbb{E}_X[(a^\top \boldsymbol{x})^2]}\sqrt{\mathbb{E}_Y[(a^\top \boldsymbol{y})^2]}} = arg\max_{w_1,w_2} \frac{a^\top \Sigma_{X,Y} b}{\sqrt{a^\top \Sigma_{X,X} a}\sqrt{b^\top \Sigma_{Y,Y} b}}$$

where $\Sigma_{X,Y}, \Sigma_{X,X} \Sigma_{Y,Y}$ are the (true) covariance and variance matrices respectively (e.g. $\Sigma_{X,Y}[i, j] = Cov[X_i, X_j] = [X_i Y_j]$ for centered random variables). All of these can be replaced by empirical data matrices in the obvious way.

## K.4 SVCCA

At a high level, SVCCA is a similarity measure of two matrices that aims in removing redundant neurons (i.e. redundant features) with the truncated SVD by keeping 0.99 of the variance and then measure the overall similarity by averaging the top $C$ CCA values.

**SV:** Given two matrices $L_1 \in \mathbb{R}^{N,D_1}, L2 \in \mathbb{R}^{N,D_2}$ (e.g. layer matrices) first reduce the effective dimensionality of the matrix via a low rank approximation $L'_1 \in \mathbb{R}^{N,D'_1}, L2' \in \mathbb{R}^{N,D'_2}$ by choosing

the top $k$ singular values that keeps 0.99 of the variance. In particular, for each layer matrix, $L_i$ keep the top $D_i'$ singular values (and vectors) such that $\sum_{j=1}^{D_i'} |\sigma_j| \geq 0.99 \sum_{j=1}^{rank(L_i)} |\sigma_j|$.

**SVCCA:** SVCCA is a statistical technique for the measuring the (linear) similarity of two sets of data sets $L_1 \in \mathbb{R}^{N,D_1}, L2 \in \mathbb{R}^{N,D_2}$ (e.g. data matrices, layer matrices) by first reducing the effective dimensionality of the matrix via a low rank approximation $L_1' \in \mathbb{R}^{N,D_1'}, L_2' \in \mathbb{R}^{N,D_2'}$ (e.g. by choosing the top $k$ singular values that keeps 0.99 of the variance) and then applying the standard empirical CCA to the resulting matrices. This is repeated $C = \min(D_1', D_2')$ times and the overall similarity of the two matrices is computed as the average CCA: $svcca = sim(L_1', L_2') = \frac{1}{C} \sum_{c=1}^{C} \rho_c$

Concretely:

1. Get the $D_i'$ components that keep 0.99 of the variance (i.e. $D_i'$ such that $\sum_{j=1}^{D_i'} |\sigma_j| \geq 0.99 \sum_{j=1}^{rank(L_i)} |\sigma_j|$ ).

2. Get the SVD: $U_1, \Sigma_1, V_1^\top = SVD(L_1)$ and $U_2, \Sigma_2, V_2^\top = SVD(L2)$

3. Then produce the SVD dimensionality reduction by $L_1' = L_1 V_1[1 : k_i] \in \mathbb{R}^{N,D_1}$ and $L_2' = L_2 V_2[1 : k] \in \mathbb{R}^{N,D_2}$ where $V_i[1 : D_i]$ gets the top $D_i$ columns of a layer matrix $i$.

4. Get the CCA of the reduced layer matrix: $[\rho_c]_{c=1}^{C} = CCA(L_1', L_2')$ where, $C = \min(D_1', D_2')$

5. Finally return the mean CCA: $svcca = \frac{1}{C} \sum_{c=1}^{C} \rho_c$, where is the k-th CCA value of the reduced layer matrix.

## K.5 PWCCA

At a high level, PWCCA was developed to increase the robustness (to noise) of SVCCA in the context of deep neural networks. In particular, Maithra et al. [35] noticed that when the performance of the neural networks stabilized, so did the set of CCA vectors (or principle neuron vectors) related to the network stabilized on the data set in question. Thus, they suggest to give higher weighting to the canonical correlation $\rho_c$ of these stable CCA vectors – in particular to the ones that are similar to the final output layer matrix, e.g. $L_1$. Note this is simpler than trying to track the stability of these CCA vectors during training and then give those higher weighting.

**PWCCA:** Formally let $L_1$ be the layer matrix $[z_d^l; \dots; z_{D_1}^l] \in \mathbb{R}^{N,D_i}$ with neuron vectors $z_d^l$ for some layer $l$. Recall that the k-th left CCA vector for layer matrix $L_1$ is defined as follows, $\tilde{x}_c = L_1 a_c = L_1(\Sigma^{-\frac{1}{2}} u_c)$ where $a_c$ is the cth CCA direction and $u_c$ is the c-th left singular value from the matrix $M = \Sigma_{L_1}^{-\frac{1}{2}} \Sigma_{L_1,L_2} \Sigma_{L_2}^{-\frac{1}{2}} = U\Lambda V^\top$. Then, PWCCA can be computed as follows:

1. Calculate the CCA vectors $\tilde{x}_c = L_1 a_c = L_1(\Sigma_{L_1}^{-\frac{1}{2}} u_c)$ and explicitly orthonormalize with Gram-Schmidt for numerical stability.

2. Compute the weight $\tilde{\alpha}_c$ of how much the layer matrix $L_1$ is account for by each CCA vector $\tilde{x}_k$ with equation $\tilde{\alpha}_c(h_c, L_1) = \sum_{c=1}^{C} |\langle \tilde{x}_c, z_c^l \rangle_{\mathbb{R}^N}|$ where $z_c^l$ is the c-th column of the layer matrix $L_1$

3. Normalize the weight indicating how much each CCA vector $h_c$ accounts for $L_1$ and denote it with, $\alpha_c(\tilde{x}_k, L_1) = \frac{\tilde{\alpha}_c(\tilde{x}_k, L_1)}{\sum_{c=1}^{C} \alpha_c(\tilde{x}_k, L_1)}$

4. Finally return the mean CCA weighted by $\alpha_c(\tilde{x}_k, L_1)$: $pwcca = \sum_{c=1}^{C} \alpha_c(\tilde{x}_k, L_1)\rho_c$ where $C = \min(D_1', D_2')$.

The original authors could have used the right CCA vectors, i.e. $\tilde{y}_c = L_2 b_k = L_2(\Sigma^{-\frac{1}{2}} v_k)$ and in fact the details of their code suggest they choose the one that would have lead to less values removed by SVD. This choice seems to already be robust to noise, as shown in [35]. Note that the CCA vectors $\tilde{x}_k, \tilde{y}_k$ are of size $\mathbb{R}^N$ and thus could be viewed as the principle neuron vectors that correlate two layers $L_1, L2$. With this view, PWCCA computes the mean CCA normalized by of the $c$ principle neuron vectors are account for the output layer matrix the most.

## K.6 CCA for CNNs

The input to CCA are two data matrices, but CNNs have intermediate representations that are 4D tensors. Therefore, some justification is needed in how to create the data matrices needed for computing CCA for CNNs. Note that it's the same reasoning for both SVCCA and PWCCA.

**Each channel as the dimensionality of the data matrix:** One option is to get the intermediate representation of size $[M, C, H, W]$ and get a layer matrix of size $[MHW, C]$. Thus, $MHW$ is the effective number of data points and the channels (or number of filters) is the effective dimensionality of the (layer) matrix. In this view, each patch of an image processed by the CNN is effectively considered a data point. This view is very natural because it also considers each filter as its own "neuron" – which seams reasonable considering that each filter uniquely responds to each stimulus (e.g., data patch). This view results in $HW$ images for every sample in the data set (or batch) of size $M$ and $C$ effective neurons.

Although the original authors suggest this metric as a good metric mainly for comparing two layers that are the same – we hypothesize it is also good for comparing different layers (as long as the effective number of data points match for the two layer matrices). The reason is that CCA tries to compute the maximum correlation of two data sets (or sets of random variables) and assumes no meaning in the ordering of the data points and assumes no process for generating each individual sample for the set of random variables, thus meaning that this metric (CCA) can be used for any two layers in a matrix. Overall, in this view, we are comparing the representation learned in each channel.

**Each activation as the dimensionality of the data matrix:** One option is to get the intermediate representation of size $[M, C, H, W]$ and get a layer matrix of size $[M, CHW]$. Thus, $M$ is the effective number of data points (which matches the number of samples in the data set or batch) and therefore each activation value is the effective dimensionality of the (layer) matrix. In this view, each activation is viewed as a neuron of size $M$ and we have $CHW$ effective neurons for each activation. The authors suggest this metric for comparing different layers (potentially at different depths). However, because CCA assumes no correspondence between the data points nor the same dimensionality in the data matrix – we hypothesize this way to define the data matrix is as valid as the previous definition for comparisons between any models at any layer. One disadvantage however is that it will often result in data matrices that are very large due to $CHW$ being very large – which results in artificially high CCA similarity values. Potential ways to deal with it are noticing that there is no correspondence between the data matrices, so a cross comparison of every data point with every other data point in CCA is possible (resulting is $O(M^2)$ comparisons for the empirical covariance matrix). Alternatively one can pool in the spatial dimensions $[H, W]$ resulting in potential smaller layer matrices e.g. of shape $[M, C]$ with a pool over the entire spatial dimension. For these reasons and the fact that we hypothesize an image patch being its own image – we prefer to interpret the number of channels as the natural way to compare CNNs so that the layer matrices results of size $[MHW, C]$.

**Subsampling of representations for channels as dimensionality:** In this section, we review the subsampling we did when comparing the representations learned in each channel, i.e. the layer matrix has size $[MHW, C]$. The effective number of data points $MHW$ will often be much larger than needed (e.g. for 16 data samples $M = 16$ and $H = W = 84$ results in $MHW = 112,896$), especially compared to the number of filters/channels (e.g. $C = 64$). Previous work [35, 38] suggest using the number of effective data points to be from 5-10 times the size of the dimensionality in a layer matrix of size $[N', D']$ that means $N' = 10D'$. Based on our reproductions of that number, we choose $N' = 20D'$ which results in $NHW = 20C$

## K.7 Centered Kernel Alignment (CKA)

At a high level, CKA is based on the insight that one can first measure the similarity between every pair of examples in each representation separately and then use the similarity structure to compute an overall similarity metric. In our case, we can treat the examples as the neuron vectors and compare all neuron vectors using some kernel function. Usually this will end in a kernel matrix of size $M, M'$ where $M$ and $M'$ are the number of examples. In our case, they would be $D, D'$ for the number of neurons of each layer matrix. Note, the layers matrices can correspond to neurons of different layers in a neural network.

**Linear CKA:** We use the linear kernel function as used in previous work [16, 24]. Given two layer matrices $X_1 \in \mathbb{R}^{N,D_l}$ and $X_2 \in \mathbb{R}^{N,D_{l'}}$ for layers $l, l'$, we compute the linear kernel $X_1^\top X_2$ to get the $D_l$ by $D_{l'}$ kernel matrix indicating the (linear) similarity per neuron vector for the two layers. Then to obtain a single distance value we compute the Frobenius norm of the kernel matrix and subtract by one after normalization:

$$d_{linearCKA}(A,B) = 1 - \frac{\|A^\top B\|_F}{\|A^\top A\|_F \|B^\top B\|_F} \tag{3}$$

Note that depending on how the examples in matrices $A, B$ are organized the cross-product could be computed with $AB^\top$ instead. Other kernel functions have been tested (e.g., the RBF kernel) for CKA but similar results are obtained, resulting in linear CKA being the most popular CKA method [16, 24] to the best of our knowledge.

### K.8  Orthogonal Procrustes Distance (OPD)

At a high level, the orthogonal Procrustes distance computes the distances between two matrices after using for the best orthogonal matrix that tries to match the two. Usually this is done after centering and dividing by the Frobenious the matrices, i.e. normalizing the matrices. In addition, previous work [16] finds that OPD is a better metric at detecting changes that matter functionally and robust against changes that do not matter.

**OPD:** Formally, the Orthogonal Procrustes Distance is the smallest distance between two matrices $X$ and $Y$ (with columns as the vectors in question) found by finding the orthogonal matrix $Q$ which most closely maps $A$ to $B$. Therefore, the OPD distance is the distance value from solving the orthogonal Procrustes problem:

$$d_{OPD}(X,Y) = \min_Q \|X - YQ\|_F \tag{4}$$

where $\|\cdot\|_F$ is the Frobenius norm. When matrices are normalized (centered and divided by their Frobenious norm) this is called the general Procrustes problem. However, the closed for equation we used is the following:

$$d'_{OPD}(X,Y) = \frac{1}{2}\left(\|X\|_F + \|Y\|_F - 2\|X^\top Y\|_*\right) \tag{5}$$

where $\|\cdot\|_*$ is the nuclear norm, i.e. the sum of singular values $\sum_i \sigma(A)_i = \|A\|_*$ . The division by 2 is to guarantee that the OPD distance is between $[0,1]$ instead of $[0,2]$. We do the standard normalization of the matrices before computing the OPD distance – by centering and dividing by the Frobenious norm of the matrix. This is done because the orthogonal matrix in the orthogonal Procrustes problem does not allow for translation or rescaling of the matrices. Therefore, this normalization enforces invariance to this type of transformations – i.e. we don't want large OPD values due to rescaling or translation (and even if present, the orthogonal matrix wouldn't be able to reflect it).

Therefore, the final equation for OPD we use is:

$$d_{OPD}(X,Y) = 1 - \|X^\top Y\|_* \tag{6}$$

**Why OPD?** We use OPD due to the findings of [16]. They find that OPD is a more robust metric (compared to SVCCA, PWCCA, and CKA) because it is *sensitive* to changes that affect real functional behavior (so it detects changes to behavior that "matter") and it's *specific* against changes that do not. As a summary, some of the evidence that they provide for this is that OPD is able to detect when 0.75 of the principal components are removed, while CKA cannot detect removal of principal components until 0.97 are removed. CCA like metrics on the other hand are not specific – even random initialization noise overwhelms the distances it reports, while OPD is more robust to this random noise. For the last point, this means that even if we compare two different layers with CCA, the noise will dominate the distance reported instead of the difference caused by comparing different layer.

### K.9  Correctly using Feature Based Distances

When comparing two layers of a neural network using two layer matrices, one needs to be careful with the number of data points (or batch size) being used. This is because metrics like CCA intrinsically

are formulated as an optimization and if the number of examples is not larger than the number of dimensionality of the examples – then the similarity can be pathologically be perfect (e.g., the distance is zero when it's actually not zero). Therefore, we follow the suggestions by the original authors of SVCCA [38] and always use at least 10 times more examples than there are features for our feature based comparisons. We call this value the *safety margin*. To illustrate this idea, we produce two random matrices and compare how the similarity (SVCCA) values varies as a function of the dimensionality of the data and the number of points. Since the two matrices are completely random, we know they should not be very similar and thus SVCCA should report a high similarity value (or low distance value). Therefore, we can see in figure 18 how as the dimensionality increases, the similarity value approaches a perfect similarity of 1.0. In figure 19 we can see how as the number of points increases, we approach a smaller similarity – closer to the true similarity for random matrices.

In general, given two matrices $X, Y \in \mathbb{R}^{M', D'}$ with the number of (effective) data points $M'$ and (effective) dimensionality (number of features) $D'$ – we want the number of points to be larger by a safety factor $s$. Formally, it must satisfy this inequality to avoid the pathological case for feature based distances:

$$D' \leq sM' \tag{7}$$

where we suggest to use $s \leq 10$ (as used in previous work [38]). Note the effective number of data points used and dimensionality can be different depending on how one reshapes the CNN tensors to produce layer matrices as explained in section K.6. For example, if one uses the channel as the dimensionality (i.e., use the image patches as an effective data point) then one has to obey the following inequality:

$$C \leq sMHW \tag{8}$$

where $M$ is the batch size, $H, W$ is the height and width of the images, and $C$ is the number of filters/channels for the current layer. This means that for a given architecture processing images of a given size that the only parameter we can change to make the above inequality true is the batch size $M$.

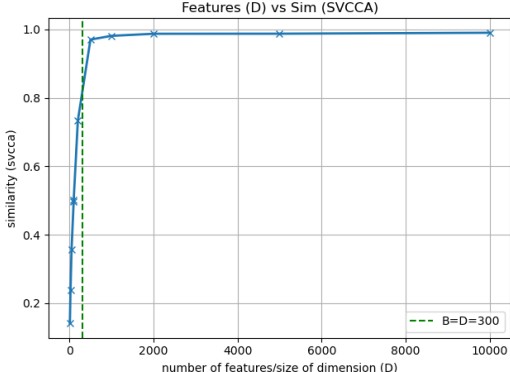

Figure 18: **Shows that as the dimensionality of a random data matrix increases – the SVCCA similarity approaches the pathological case by falsely reports the similarity is perfect.** The green line indicates when the number of examples and dimensionality are equal (and equal to 300). D denotes the dimensionality of the simulated data and B the size of the batch size/number of points.

## L   A Statistical Decision view of the differences between Supervised Learning and Meta-learning

Recent work in meta-learning implies that feature-reuse might be all we need to solve modern few-shot learning benchmarks [43]. However, what it also reveals is our poor understanding of meta-learning algorithms. Therefore, in this section, we take the most foundational perspective to formulate and analyze meta-learning algorithms by analyzing them from an optimal statistical decision theory perspective?

We hope that this can help clarify the results from [43] and therefore help meta-learning researchers design better meta-learning benchmarks and meta-learning algorithms.

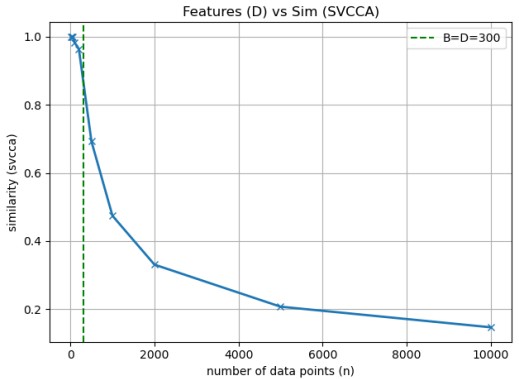

Figure 19: **Show how to avoid the pathological case when using feature based similarities by increasing the number of data points (or batch size).** In particular, as the number of data points in two random data matrix increases – the true similarity approaches the true low similarity value. The green line indicates when the number of examples and dimensionality are equal (and equal to 300). D denotes the dimensionality of the simulated data and B the size of the batch size/number of points.

### L.1 Supervised Meta-Learning problem set-up

In this section, we introduce the notation for supervised meta-learning. Intuitively, we seek to find a function that minimize the expected risk over tasks and the data in the tasks. To formalize it, we will use three formulations:

**Monolithic meta-learner**: for a monolithic decision rule $g$ (or meta-learner), we want to find the optimal $g$ by minimizing the *supervised meta-learning expected risk*:

$$R_{Mono}(g) = \mathbb{E}_{\tau \sim p(\tau)} \mathbb{E}_{x,y \sim p(x,y|\tau)} \left[ l(g(x,\tau), y) \right] \tag{9}$$

where $g$ is a single monolithic function, $p(\tau)$ is the true but unknown distribution of tasks, $p(x, y \mid \tau)$ is the true, but unknown distribution of data pair given a task $\tau$ and $(x, y)$ is the data pair of input and target value sampled from a task.

**Meta-learned meta-learner**: for a meta-learned decision rule we usually have an adaptation rule $A$ (e.g. SGD in MAML) and a function approximator $h$ (e.g. a neural network) and minimize the follow over both:

$$R_{ML}(A, h) = \mathbb{E}_{\tau \sim p(\tau)} \mathbb{E}_{x,y \sim p(x,y|\tau)} \left[ l(A(h, \tau)(x), y) \right] \tag{10}$$

$p(\tau)$ is the true but unknown distribution of tasks, $p(x, y \mid \tau)$ is the true, but unknown distribution of data pair given a task $\tau$ and $(x, y)$ is the data pair of input and target value sampled from a task.

**Fixed representation meta-learner without adaptation**: one can also solve 9 using a single decision rule $f$ that does not take the task $\tau$ as input as follows:

$$R_{SL}(f) = \mathbb{E}_{\tau \sim p(\tau)} \mathbb{E}_{x,y \sim p(x,y|\tau)} \left[ l(f(x), y) \right] \tag{11}$$

where $f$ is a function to be adapted (e.g. a neural network), $p(\tau)$ is the true but unknown distribution of tasks, $p(x, y \mid \tau)$ is the true, but unknown distribution of data pair given a task $\tau$ and $(x, y)$ is the data pair of input and target value sampled from a task.

**Fixed representation meta-learner with a final adaptation layer**: one can also solve 9 using a single feature extractor $g$ that does not take the task $\tau$ as input with a feature extractor $g$:

$$R_{SLA}(f, g) = \mathbb{E}_{\tau \sim p(\tau)} \mathbb{E}_{x,y \sim p(x,y|\tau)} \left[ l((f(\tau) \circ g)(x), y) \right] \tag{12}$$

where $g$ is the feature extractor from the raw inputs (e.g. a neural network), $f$ the final layer adapted (e.g. a linear layer), $p(\tau)$ is the true but unknown distribution of tasks, $p(x, y \mid \tau)$ is the true, but unknown distribution of data pair given a task $\tau$ and $(x, y)$ is the data pair of input and target value sampled from a task.

*Remark* L.1. Note that in practice, the meta-learner does not usually take the full task $\tau$ as input, but instead a train and test set (often referred to as support set and query set) sampled from the task $\tau$.

The goal of this work is to clarify the difference between 11 and 10 under the framework of statistical decision theory. Arguably the most important comparison between 10 and 12 is left for future work.

 **L.2   Main Result: Difference between the Supervised Learned and Meta-learned decision rule**

1017 The proof sketch is as follows: we first show the optimal decision rules for both supervised learning
1018 and meta-learning when minimizing the expected meta-risk from equations 11 and 10 and then
1019 highlight that the main difference between them is that the meta-learned solution can act optimally if
1020 it identifies the task $\tau$ while the supervised learned solution has no capabilities of this since it learns
1021 an average based on tasks instead.

1022 **Theorem L.2.** *The minimizer to equation 10 is:*

$$A(h, \tau)(x) = \bar{y}^*_{y|x,\tau} = \mathbb{E}_{y \sim p(y|x,\tau)}[y] \tag{13}$$

1023 *where $\bar{y}^*_{y|x,\tau} = \mathbb{E}_{y \sim p(y|x,\tau)}[y]$ and l is the squared loss $l(\hat{y}, y) = (\hat{y} - y)^2$.*

*Proof.* The proof is the same as the standard decision rule textbook proof but instead of minimizing
it point-wise w.r.t. $x$ we minimize it point-wise w.r.t. $(x, \tau)$. In particular, we have:

$$R_{ML}(A, h) = \mathbb{E}_{\tau \sim p(\tau)}\mathbb{E}_{x,y \sim p(x,y|\tau)}[l(A(h, \tau)(x), y)]$$

$$\min_{A,h} \mathbb{E}_{\tau \sim p(\tau)}\mathbb{E}_{x \sim p(x|\tau)}\mathbb{E}_{y \sim p(y|x,\tau)}\left[(A(h, \tau)(x) - y)^2\right]$$

without loss of generality (WLOG) and for clarity of exposition consider the special case for discrete
variables:

$$\min_{A,h} \sum_\tau p(\tau) \sum_x p(x \mid \tau)\mathbb{E}_{y \sim p(y|x,\tau)}\left[(A(h, \tau)(x) - y)^2\right]$$

At this point we notice we can minimize the above point-wise w.r.t $(x, \tau)$ and ignore $h$. To do that,
take the derivative of $R(A, h)$ with respect to $A(h, \tau)(x)$ because that $A(h, \tau)(x) \in \mathbb{R}$ and set it to
zero:

$$\frac{d}{dA(h,\tau)(x)}\mathbb{E}_{y \sim p(y|x,\tau)}\left[(A(h, \tau)(x) - y)^2\right] = 0$$

$$\mathbb{E}_{y \sim p(y|x,\tau)}[(A(h, \tau)(x) - y)] = 0$$

$$\mathbb{E}_{y \sim p(y|x,\tau)}[(A(h, \tau)(x)] = \mathbb{E}_{y \sim p(y|x,\tau)}[y]$$

$$A(h, \tau)(x) = \mathbb{E}_{y \sim p(y|x,\tau)}[y] = \bar{y}^*_{y|x,\tau}$$

1024 as desired. □

1025 **Corollary L.3.** *For a monolithic meta-learner defined in section L.1 the solution to the meta*
1026 *supervised learning problem is the same as in equation 13 for the squared loss $l(\hat{y}, y) = (\hat{y} - y)^2$ i.e.*
1027 $g(\tau, x) = \bar{y}^*_{y|x,\tau} = \mathbb{E}_{y \sim p(y|x,\tau)}[y].$

1028 *Proof.* Proof is trivial, replace $A(h, \tau)(x)$ with $g(\tau, x)$ since $h$ is not used. In this case, there is
1029 no difference with having an adaptation rule $A$ equipped with another function $h$ and a monolithic
1030 meta-learner $g$. □

1031 **Theorem L.4.** *The minimizer to equation 11:*

$$f(x) = \mathbb{E}_{\tau \sim p(\tau|x)}\left[\bar{y}^*_{y|x,\tau}\right] \tag{14}$$

1032 *where $\bar{y}^*_{y|x,\tau} = \mathbb{E}_{y \sim p(y|x,\tau)}[y]$ and l is the squared loss $l(\hat{y}, y) = (\hat{y} - y)^2$.*

*Proof.* WLOG, consider the minimizer of equation 11 in the discrete case. In particular, we have:

$$R_{SL}(f) = \mathbb{E}_{\tau \sim p(\tau)}\mathbb{E}_{x,y \sim p(x,y|\tau)}[l(f(x), y)]$$

$$\min_f \mathbb{E}_{\tau \sim p(\tau)}\mathbb{E}_{x \sim p(x|\tau)}\mathbb{E}_{y \sim p(y|x,\tau)}\left[(f(x) - y)^2\right]$$

$$\min_f \sum_x \mathbb{E}_{\tau \sim p(\tau)}p(x \mid \tau)\mathbb{E}_{y \sim p(y|x,\tau)}\left[(f(x) - y)^2\right]$$

Note we can minimize the above point-wise w.r.t. $x$ only (and not also w.r.t. $\tau$ as we did in proof
L.2). Thus, we have want:

$$f(x) = \min_{f(x) \in \mathbb{R}}\mathbb{E}_{\tau \sim p(\tau)}p(x \mid \tau)\mathbb{E}_{y \sim p(y|x,\tau)}\left[(f(x) - y)^2\right]$$

at this point it is interesting to observe the disadvantage of supervised learning methods with fixed functions without dependence on the task is that they are forced to consider all task $\tau$ at once. We proceed to take derivatives as in proof L.2 but with this objective:

$$\mathbb{E}_{\tau\sim p(\tau)}p(x\mid\tau)\mathbb{E}_{y\sim p(y\mid x,\tau)}\left[(f(x)-y)^2\right]$$

$$\frac{d}{df(x)}\mathbb{E}_{\tau\sim p(\tau)}p(x\mid\tau)\mathbb{E}_{y\sim p(y\mid x,\tau)}\left[(f(x)-y)^2\right]=0$$

$$\mathbb{E}_{\tau\sim p(\tau)}p(x\mid\tau)\mathbb{E}_{y\sim p(y\mid x,\tau)}\left[f(x)\right]=\mathbb{E}_{\tau\sim p(\tau)}p(x\mid\tau)\mathbb{E}_{y\sim p(y\mid x,\tau)}\left[y\right]$$

$$f(x)\mathbb{E}_{\tau\sim p(\tau)}\left[p(x\mid\tau)\right]=\mathbb{E}_{\tau\sim p(\tau)}p(x\mid\tau)\mathbb{E}_{y\sim p(y\mid x,\tau)}\left[y\right]$$

$$f(x)=\mathbb{E}_{\tau\sim p(\tau)}\left[\frac{p(x\mid\tau)}{\mathbb{E}_{\tau\sim p(\tau)}\left[p(x\mid\tau)\right]}\mathbb{E}_{y\sim p(y\mid x,\tau)}\left[y\right]\right] \qquad (15)$$

We proceed by noticing that $\mathbb{E}_{\tau\sim p(\tau)}\left[p(x\mid\tau)\right]=p(x)$, thus:

$$f(x)=\mathbb{E}_{\tau\sim p(\tau)}\left[\frac{p(x\mid\tau)}{p(x)}\mathbb{E}_{y\sim p(y\mid x,\tau)}\left[y\right]\right]$$

$$f(x)=\sum_{\tau}p(\tau)\frac{p(x\mid\tau)}{p(x)}\left[\mathbb{E}_{y\sim p(y\mid x,\tau)}\left[y\right]\right]$$

$$f(x)=\sum_{\tau}\frac{p(\tau)}{p(x)}\frac{p(x,\tau)}{p(\tau)}\left[\mathbb{E}_{y\sim p(y\mid x,\tau)}\left[y\right]\right]$$

$$f(x)=\sum_{\tau}\frac{p(x,\tau)}{p(x)}\left[\mathbb{E}_{y\sim p(y\mid x,\tau)}\left[y\right]\right]$$

$$f(x)=\sum_{\tau}p(x\mid\tau)\left[\mathbb{E}_{y\sim p(y\mid x,\tau)}\left[y\right]\right]$$

$$f(x)=\mathbb{E}_{\tau\sim p(x\mid\tau)}\left[\mathbb{E}_{y\sim p(y\mid x,\tau)}y\right]$$

$$f(x)=\mathbb{E}_{\tau\sim p(x\mid\tau)}\left[\bar{y}^*_{y\mid x,\tau}\right]$$

as required by the rightmost RHS of equation 14. □

**Theorem L.5.** *The minimizer in equation 14 reduces to an expectation only over w.r.t. $p(\tau)$ of $\bar{y}^*_{y\mid x,\tau}$ under benchmarks that are balanced. Formally*

$$f(x)=\mathbb{E}_{\tau\sim p(\tau)}\left[\bar{y}^*_{y\mid x,\tau}\right]=\mathbb{E}_{\tau\sim p(\tau)}\left[\mathbb{E}_{y\sim p(y\mid x,\tau)}\left[y\right]\right] \qquad (16)$$

*where $\bar{y}^*_{y\mid x,\tau}=\mathbb{E}_{y\sim p(y\mid x,\tau)}\left[y\right]$ and under assumption A1: $p(x\mid\tau)$ is a constant, i.e. $p(x\mid\tau)=k_{XT}\in\mathbb{R},\forall x\in X,\forall\tau\in T$ and $l$ is the squared loss $l(\hat{y},y)=(\hat{y}-y)^2$.*

*Proof.* Recall equation 14:

$$f(x)=\mathbb{E}_{\tau\sim p(\tau\mid x)}\left[\bar{y}^*_{y\mid x,\tau}\right]$$

due to Bayes's rule we have $p(\tau\mid x)=\frac{p(\tau)p(x\mid\tau)}{p(x)}$ and equation 14 can be re-written as follows:

$$f(x)=\mathbb{E}_{\tau\sim p(\tau)}\left[\frac{p(x\mid\tau)}{p(x)}\bar{y}^*_{y\mid x,\tau}\right]$$

under assumption A1 we have that $p(x\mid\tau)$ does not depend on as a function of $x$ or $\tau$. Thus, we have:

$$p(x)=\sum_{\tau}p(\tau)p(x\mid\tau)=p(x\mid\tau)\sum_{\tau}p(\tau)=p(x\mid\tau)$$

Thus we have:

$$f(x)=\mathbb{E}_{\tau\sim p(\tau)}\left[\frac{p(x)}{p(x)}\bar{y}^*_{y\mid x,\tau}\right]$$

$$f(x)=\mathbb{E}_{\tau\sim p(\tau)}\left[\bar{y}^*_{y\mid x,\tau}\right]$$

as required. □

*Remark* L.6. Note that assumption A1 holds for the common MiniImagenet few-shot learning data set, where $p(x \mid \tau) = \frac{1}{600}$.

*Remark* L.7. In addition, because all classes are equally likely (e.g. $p(class) = \frac{1}{64}$ for the meta-train set) we have $p(\tau)$ is the same constant independent of the task $\tau$. Proof in the appendix, lemma L.8.

**Theorem L.8.** *If the tasks are equally likely, then equation 16 becomes an average over conditional predictions over all tasks. Formally, if $p(\tau) = \frac{1}{T}$ then equation 16 becomes:*

$$f(x) = \frac{1}{T} \sum_{\tau} \bar{y}^*_{y|x,\tau} \tag{17}$$

*under the squared loss $l(\hat{y}, y) = (\hat{y} - y)^2$.*

*Proof.* Since $f(x) = \mathbb{E}_{\tau \sim p(\tau)} \left[ \bar{y}^*_{y|x,\tau} \right]$ then, plugging $p(\tau) = \frac{1}{T}$ completes proof. $\square$

*Remark* L.9. It is interesting to note that without adaptation or dependence on the task $\tau$ being solved, the supervised learned meta-learner is suboptimal compared to the meta-learned solution. The proof is simple, and it follows because the meta-learned decision rule was chosen to minimize each term individually, but the supervised learned decision is not of that form. Proof in appendix L.11. Unfortunately, note that this does not necessarily apply to previous work [43].

*Remark* L.10. Note that remark L.9 does not apply to work [43] because that work does depend on a task $\tau$ during meta-test time by adapting the final layer even if the representation is fixed.

*Remark* L.11. The supervised learning decision rule is suboptimal compared to the meta-learned decision rule.

### L.3 The supervised Learning Solution is equivalent to the Meta-Learning solution when there is low task diversity

**Sketch argument:** The main idea is that because all tasks are very similar (task diversity is low) – it essentially means that $\tau$ is not truly an input to the adaptation rule or monolithic meta-learner). Equivalently, the problem is essentially a single task problem, so the task is implicitly an input to any method used. Therefore, since the task conditioning does not exist, then the optimization problem is the same for the meta-learned solution and when there is a fixed supervised learning feature extractor.

**Theorem L.12.** *Assume $\tau_1 = \tau_2$ for any tasks in $T$ and the data sets are balanced (i.e. same number of images $x$ for each task). Then we have the meta-learned solution is the same as the supervised learning solution with shared embeddings: $f_{sl}(x) = A(f_{ml}, \tau)(x)$.*

*Proof.* Consider the optimization problem, for supervised learning:

$$\min_{A,h} \mathbb{E}_{\tau \sim p(\tau)} \mathbb{E}_{x \sim p(x|\tau)} \mathbb{E}_{y \sim p(y|x,\tau)} \left[ (A(h, \tau)(x) - y)^2 \right]$$

If every pair of tasks is equal, it means their distributions are equal $p(x, y \mid \tau) = p(x, y)$ (meaning $\tau$ can be ignored). Thus, the solution to the supervised learning problem is: $f_{sl}(x) = \mathbb{E}_{\tau} \mathbb{E}_{p(x,y)}[y] = \mathbb{E}_{p(x,y)}[y] = y^*_{|x}$. Now for the meta-learning problem we have: $A(f_{ml}, \tau)(x) = y^*_{|x,\tau} = \mathbb{E}_{y \sim p(y|x,\tau)}[y]$ but due to every pair of tasks being equal means $p(x, y \mid \tau) = p(x, y)$ (i.e. all task share the same distributions) we have: $A(f_{ml}, \tau)(x) = \mathbb{E}_{y \sim p(y|x,\tau)}[y] = \mathbb{E}_{y \sim p(y|x)}[y] = y^*_{|x}$ which is the same as the solution as in $f_{sl}$. Thus $f_{sl}(x) = A(f_{ml}, \tau)(x)$. $\square$

*Remark* L.13. Proofs were presented in the discrete case clarity, but it is trivial to expand them to the continuous case – e.g., using integrals instead of summations.

## M  Summary of Compute Required

We used an internal compute cluster with wide varied of GPUs. We used Titan X GPUs for most five layer CNN experiments. We used A40 and dgx-A100 GPUs for Resnet12 experiments, with 48 GB and 40 GB GPU memory respectively. We did notice that the Resnet12 architecture we used from previous work [43] required more memory than Resnet18 and Resnet34 used in Task2Vec [4]. By requiring more memory, we mean we did not have many memory out of bound issues with

Resnet18/Resnet34 but did have memory issues with Resnet12. In addition, our episodic meta-learning training for MAML used Learn2Learn's [5] distributed training to speed up experiments. Experiments took 1-2 weeks with MAML in a single GPU to potentially 2-3 days with multiple GPUs (we used 2, 4 to 8 GPUs depending on availability). For synthetic experiments we used Titan X GPUs with 16GB of GPU memory. Experiments took around 1-2 days on average with a single GPU. For more precision check the experimental details section D.