# OpenReview forum: "The Curse of Low Task Diversity: On the Failure of Transfer Learning to Outperform MAML and their Empirical Equivalence"
_NeurIPS.cc/2022/Conference — NeurIPS 2022 Submitted_

### Official Review · Reviewer_JSFh · 2022-06-19

**Rating:** 3
**Confidence:** 4
**Soundness:** 3 good
**Presentation:** 2 fair
**Contribution:** 2 fair

**Summary:**

The paper analyses some of the earlier claims about a simple transfer learning baseline outperforming SOTA meta-learning methods through the lens of task diversity. The paper posits, that with a fair comparison (e.g. same architecture, optimizer etc. ) MAML and transfer learning methods are statistically similar when the meta-training task diversity is low.

**Questions:**

The questions are stated in Strengths and Weaknesses.

**Limitations:**

While the paper tackles an interesting and important question on when to use transfer learning vs. meta-learning, the paper falls short of execution in its current form. I would suggest to authors to revisit the paper, look into the comments and make the draft stronger with a) more experiments on the task diversity metric; (b) insights and experiments on how to link task diversity with few-shot methods; c) extending beyond MAML and testing more meta-learners.

**Strengths And Weaknesses:**

Strengths:
	- The paper tackles an important question on the differences b/w transfer learning and meta-learning methods. It's an important question for the few-shot community whose answer is still not clear except for certain empirical insights. Hence, it's a good problem to study.


While the problem being studied is important, the paper lacks in execution and does not have solid insights or takeaways.

- The task diversity metric is not novel, but that is fine if more analysis is shown on why such a metric captures task diversity well. The current diversity metric is computed only on miniImagenet and CIFAR which are toy datasets for few-shot learning currently. It would be a good idea to take a wide range of datasets (e.g. MetaDataset, ORBIT, VTAB) and show how the task diversity metric plays out. I feel the current analysis and subsequent insights are incomplete.

-  The empirical results state that under a fair condition, when task diversity is low: MAML and Supervised learning methods are equivalent. What is only MAML chosen for the analysis? It is quite far from the state-of-the-art meta-learning methods. It would be a good idea to choose more meta-learning methods to provide a complete picture. While there exists plethora of methods, some technically simple, but well performing methods could be chosen (e.g., R2D2, MetaOptNet, FEAT).

- There are no empirical results when the task diversity is high. It would be good to run such analysis on more real-world datasets.
I couldn't understand the clear takeaway or message from the paper except for certain empirical insights. From a practical standpoint, it would be good to link the effectiveness of task diversity to how a method might be chosen. For e.g., if the task diversity has a direct correlation with certain methods, just computing the task diversity can help choose from a variety of meta-learners.

---

> ### Author Response · Authors · 2022-08-01
> **The focus of this work is an in depth study of the *low* diversity regime**
>
> Responses to the weaknesses:
>
> Bullet point 1:
> Our diversity coefficient accurately represents diversity because Task2Vec had extensive experiments showing that their task vectors capture taxonomic and semantic properties of tasks and labels. Therefore, they concluded that Task2Vec was a good vectorial representation of tasks. Therefore, the computation of diversity of task is correctly computed. In other words, the same strengths are transitively inherited by our method. There might be a concern for outlier tasks due to the averaging operation, but we addressed that in the supplementary section by plotting the heat maps of the individual pair-wise distance of tasks. In addition, for synthetic tasks, the *true* diversity can be computed because we have the true (Hellinger) distance between distributions. Therefore, we show in the supplementary section that the Task2Vec diversity and Hellinger diversity correlate well with an R=0.99 in figure 11.
> The reviewer mentioned Meta-Dataset which is an interesting point. The main issue with Meta-Dataset is that preliminary results on its derivatives show that it is a medium/high diversity data set. This is the main reason we deliberately excluded it in our analysis, since it’s not the low diversity regime we are studying (see title of paper). Since our focus was to 1. analyze if previous work claiming USL was better than MAML were true and 2. if they were not why – which we found was on the low diversity regime. This is why we focused on an in depth analysis of the low diversity regime, and we decided that Meta-Dataset was out of scope since the low diversity regime is already rich in analysis & novel, as we believe our paper shows.
> If we included Meta-Dataset, showed it was medium/high diversity, and showed the difference between MAML and USL is none zero (which is what we would expect & observing in preliminary results) the only place I see it going is in the supplementary section. Meta-data set simply isn’t the type of data set that explains our observations or puts previous work in context with a fair comparison. Or we are sincerely curious and respectfully ask, how would JSFh think it would help to include results from medium/high diversity data sets?
>
>
> Bullet point 2:
> The main reason we chose MAML is because of MAML’s simplicity & richness of analysis we already had. We could have provided more methods to do the analysis but we opted for other types of analysis we thought were more important. For example, Figure 3 compares the difference between MAML vs USL in the low diversity regime using as the size of the model increased. We initially hypothesized that for small models the meta-learned model (i.e. MAML) would make more of a difference against USL – but our work shows it did not, and they still performed similarly. Simplicity has an additional technical advantage: it makes the comparison most fair since there are less complicated moving parts. Initially, we tried to do the comparison by controlling model complexity but the additional SGD step in MAML makes it unclear how to quantitatively include that into model complexity. Instead, we opted for the simplest algorithms with the same settings and once the algorithm could not improve further (e.g. have converged or had zero train error) we proceeded to do the analysis.
> However, we do not disagree it would be interesting to do more analysis with other methods but given the time window, we are unsure if we could train enough models to meet the request. If yes, by when would they be needed?
>
> Bullet point 3:
> Bullet point 3 is the trickiest to address, since this was done by design. We discovered that the low diversity regime was already rich in analysis and provided an interesting insight – since we found the setting where in fact MAML was not worse than USL – contrary to what was reported previously *in the same benchmarks*.
> However, we actually did provide some analysis of the high diversity regime in figure 4 – where we reproduced the overperformance of USL against MAML from a problem-centric perspective.
> We do admit that more realistic data sets were missing, but preliminary results shows they do not exhibit low diversity regime. Therefore, we decided it was worthy of a separate in-depth analysis like the one we did here.
> We will address the comment on “the practical standpoint” of our method. Our method’s most practical impact is directly for AI researchers. In particular, we suggest the following paradigm shift:
> Providing a new tool for the creation of data sets and moving away from only making them larger and larger – instead, we suggest analyzing intrinsic properties of the data itself.
> A novel way to report meta-learning results for researchers. We believe that previous publications were misleading – especially because methods are compared without any control for variables like neural network backbone. People believe meta-learning might not work now – do we really know this?

---

### Official Review · Reviewer_PmXX · 2022-07-07

**Rating:** 4
**Confidence:** 4
**Soundness:** 2 fair
**Presentation:** 1 poor
**Contribution:** 2 fair

**Summary:**

The aim of the paper is to empirically compare the performance of transfer-learning (fine-tuning last fully connected layer in a pre-trained neural network) and meta-learning (and in particular, MAML - an instance of meta-learning). The two learning approaches of interest are investigated in different points of views: intrinsic diversity of the dataset used, similarity between their features at some imtermediate layers of the neural networks used and different model sizes. The empirical results show that MAML and fine-tuning a pre-trained model are equivalent in terms of performance.

**Questions:**

The proposed *diversity coefficient* based on an average distance is not very convincing. I wonder if the diversity of a dataset can be calculated informatically. What I mean is that given a set of vectors representing tasks (obtained from Task2Vec), could we estimate the task distribution and explicitly calculate the entropy of that distribution, or implicitly calculate the entropy from those vectors? The reason for entropy is that the higher the entropy, the more informative that task distribution, which, to me, means the higher diversity.

I have a concern about comparison between meta-learning and transfer-learning. Since both are using the blackbox approach, calculating the feature similarity of hidden layers might not be a good idea. The reason is that if two things are similar at the output, it does not mean that their components must be the same. Since their classification accuracy is similar, I suggest to look at the model complexity and we can base on the Occam's razor principle to see which method is better. Or, could we use different evaluation metric, such as calibration error, to see how difference the two methods are?

**Limitations:**

The current form of the paper is quite limited since it is only applicable for one instance of meta-learning (MAML).

**Strengths And Weaknesses:**

**Strengths**

The major strength of the paper is to study the difference between meta-learning and transfer-learning. This is an interesting research direction to understand which method performs better in which setting. Also, as stated at the end of the introduction section, it provides another point of view in terms of evaluation, not just simply making larger and larger datasets.

Another strength of the paper is to quantify the characteristic of tasks sampled according to the episode setting. The newly-proposed metric, *diversity coefficient* , is based on the representation of tasks, known as Task2Vec, to calculate the cosine similarity between tasks belonging to the same dataset. This quantifies the "information" of the dataset, providing insights into the performances of the model of interest.

One more strength is that the paper is gone through a thorough review of the literature with an extended section in the literature.

---

**Weaknesses**

Firstly, the paper lacks of a coherence to connect all the contributions together. As I understand, the main goal is to show that meta-learning and transfer learning share similar performances even when the dataset used is diversity enough or not. However, the presentation and writing leads to a hard time for me to get to the point.

Secondly, the newly-proposed metric, known as *diversity coefficient*, is a straight-forward extension from Task2Vec, which heavily relies on the *probe* network used. Thus, I believe that using a single network is not enough, but we might need an ensemble of networks to marginalize out the bias of *probe* networks. In addition, the proposed metric might not work well in the present of outlier tasks since it is based on the expected (or average) similarity distance. Furthermore, as shown in (Dhillon et. al 2020, Figure 1) and (Nguyen et. al 2021, Figure 1), the accuracy of different tasks evaluated by the same model varies significantly. Thus, simply picking a small number of tasks, e.g. 500 tasks as mentioned in Table 1, to calculate the metric is inadequate. And since the number of tasks formed from a dataset is very large, e.g. there are $n = \binom{64}{5} = 7,624,512$ 5-way classification tasks from the training set of mini-ImageNet, the number of distances calculate would be $\mathcal{O}(n^{2})$, making this metric intractable when there are more classes. Of course, we might not need to take all tasks into account, but only consider the set of *core* tasks. However, finding that set is another problem.

Finally, the evaluation to compare meta-learning and transfer-learning is based on classification accuracy on two datasets only. I believe that this might be insufficient to have a clear conclusion.

**References**
- Guneet S Dhillon, Pratik Chaudhari, Avinash Ravichandran, and Stefano Soatto. "A baseline for few-shot image classification". In *International Conference on Learning Representations*, 2020.
- Cuong C. Nguyen, Thanh-Toan Do, and Gustavo Carneiro. "Probabilistic task modelling for meta-learning". In *Uncertainty in Artificial Intelligence*, 2021.

---

> ### Author Response · Authors · 2022-08-01
> **We respectfully disagree accuracy is not sufficient because historically that is what ML research uses and we want an objective way to fairly compare meta-learners**
>
> We are happy to add a conclusion that makes the connection explicit of all the points into one coherent argument. The coherent argument is stated in the contribution list but we can emphasize it in a conclusion section. In one sentence: we provide extensive and diverse evidence to back up our main claim that under low diversity, MAML and USL are similar.
>
> It is true that the diversity coefficient depends on the probe network. However, we did address this issue by providing 4 different probe networks in Table 1. This approach is arguably superior to an ensemble – since it doesn’t combine all diversities into a single number.
>
> We did take precautions against outlier tasks. We reported confidence intervals and heat maps for individual task distances in Figures 13, 14, and 12. This should address this issue since we can clearly see the homogeneity of tasks.
> The accuracy indeed can vary a lot, and that is why we provided 95%-confidence intervals with sufficient data points for all our experiments. Previous work used 600 [8] and we followed a similar value.
>
> We want to respond to PmXX’s raised point on using “only” 500 tasks. The main issue is that combinatorial arguments are insufficient to calculate the *true* number of different tasks. This is why we developed the diversity coefficient. Using the formula Choose(L, n) = (eL/n)^n completely ignore the structure of the data and therefore massively overestimates the number of tasks. Therefore, the solution we choose to estimate if 500 is enough is being that number on:
> 1. it’s close to previous work [8]
> 2. we noticed the confidence intervals shrunk sufficiently
> 3. the heat maps in the supplementary section Figure 13, 14 showed the lack of variation & outliers.
> However, one alternative solution is to treat the problem compositionally. Since tasks are made using the say L=64 labels in the data sets, then we can instead use those labels as “the core tasks”. Now the run time decreases from exponential O(eL/n)^n  to squared O(L^2).
> Note that we did not do this because we believed the provided reasons were sufficient but it’s easy to compute if the reviewer requires them.
>
> We are curious why reviewer PmXX thinks accuracy is not a valid way to compare models. The community of machine learning has focused for many years on using such metrics and thus it’s unclear why a different one is better, e.g. we aren’t using medical images so type errors lack context. In addition, accuracy is the stated metric in the problem statement. Accuracy is great because it is “absolute” in the sense that it’s not an arbitrary number like the type of errors regression or losses give. Even normalized L2 errors like NED or R^2 aren’t as easy to compare to accuracy. Overall we don’t fully appreciate the issues, especially since we are responding to the current trend on claiming USL is better than MAML and those claims are made wrt accuracy.
>
> However, we actually *do* have at least one metric that isn’t accurate to show the similarity of MAML and USL. This is the similarity of the classification layer using SVCCA, PWCCA, LINKCKA, and OPD in figure 2. In particular, there we show that similarity increases at the final layer. This agrees with the accuracy analysis.
>
> We did not use information theory based metrics because 1. estimating entropy in our scenario is hard and open (see Wikipedia entry for entropy estimation). 2. our empirical experiments were based on the theory in the supplementary section K of the paper that depends on distance of tasks. To expand on this final point, the hypothesis is that if there is no difference between tasks then there is no need to learn to adapt – which is what we tested inspired by the theory in section K. Finally we don’t think this point is enough to reject our work because Task2Vec is grounded on extensive experiments, and in addition, we showed that our Task2Vec diversity correlates with the *true* Hellinger distance of tasks in the supplementary section H figure 11. Therefore, the Hellinger connection is stronger, since Hellinger can be connected to KL and other divergence metrics.
>
>
> My apologies if I don’t fully understand your final point – but we do not want to misrepresent your suggestions. We believe the main objective is that you want us to change the analysis away from accuracy. I definitely want to understand this deeper because as I expressed earlier I don’t see a serious issue with it. I do understand that two models are not the same if their output is the same, and we don’t claim that. In fact, we point out that the feature layers are different, although the accuracy is very similar, and pointed it out as a fascinating point in figure 2. The main contribution is that the two models are indistinguishable from a performance perspective, which is contrary to what other works claim – e.g. [8] . We are happy to clarify this in the paper if this is the issue.

---

> > ### Author Response · Authors · 2022-08-01
> > **Our experimental setting approximates using similar mode complexity**
> >
> > A comment on your suggestion with respect to model complexity. One question is what model complexity use that takes into account the gradient steps in MAML? This is one of the reasons we didn’t take the model complexity route since it seems tricky to take an arbitrary adaptation meta-learner step into account. Even if we could do that I’d suggest doing the reverse of your suggestion: use the model complexity to ensure a fair comparison and then compare their test performance (which is what we approximately did). If we do what you suggest i.e. achieve a certain accuracy and then choose the model that is “simpler” according to some model complexity measure – then we are not objective in what “best” since that makes it mean “simpler”. We instead want a fair comparison and let the performance speak for itself – which is why we followed previous work with test accuracy.

---

> > ### Author Response · Authors · 2022-08-01
> > **Testing and training conditions match -- so test accuracy accurately represents the difference in model**
> >
> > Final comment on why we think accuracy is a good metric, besides all the previously mentioned arguments. In addition, we suspect the reviewer might think that, even though two models have the same accuracy performance, they might actually be quite different at test time. This does not apply for any of the benchmarks we tried because the task distribution is the same at train and at test time. For example, for MiniImagenet all tasks were sampled using images from Imagenet (natural images) and processed the same way. So the sampling of tasks at train and test time match. Same with our synthetic experiments. Comparing functions is hard and there is no computationally feasible complete solution. Evaluating accuracy differences seems sufficient given the stated goals & how it’s been done historically.

---

> > ### Author Response · Authors · 2022-08-02
> > **Why 500 tasks is enough to compute the diversity coefficient**
> >
> > We’d like to convince you that our 500 samples are enough to make strong statistical inferences about the population -- even if it’s as large as 7,624,512. If we assume the distribution of the data is Gaussian, then we expect to see a single mode with an approximate bell curve. If we plot the histogram of task pair distances of the 500 tasks and see this then we can infer our Gaussian assumption is approximately correct. Given that we do see that in figure X, then we can infer our assumption is approximately correct. This implies we can make strong statistical assumptions about the population – in particular, that we have a good estimate of the diversity coefficient using 500 samples.
> >
> > We update the supplementary material, section J.2 to address this and show all 15 histograms to back up our argument.

---

> > ### Comment · Reviewer_PmXX · 2022-08-08
> > **Comments from the rebuttal**
> >
> > Thank you the authros for addressing my concerns raised in the initial review. However, I am not satisfied with the reply from the authors. Please see my comments below.
> >
> > > Providing different results using different probe networks is more superior than an emsemble approach
> >
> > This is quite arguable and I will agree if one can show that the 4 probe networks are not converged to the same point or result in the same bias. The reason I asked for a probabilistic modelling approach is to consider the whole distribution of models, and hence, we can marginalize the effect of model biasing toward to the data, especially in case of point estimation. If the distribution of models are used, the result would help to shape a more convincing observation and conclusion.
> >
> > > The accuracy indeed can vary a lot, and that is why we provided 95%-confidence intervals with sufficient data points for all our experiments. Previous work used 600 [8] and we followed a similar value.
> >
> > I still stand my point since reporting the result on a subset of data (500 data points) with confident interval in this case does not provide a good summary on the accuracy result. In addition, previous work used 600 tasks does not mean that they did the correct thing (as shown in the two papers I refered in my initial review).
> >
> > > accuracy is not a valid way to compare models
> >
> > I did not say that accuracy is an invalid metric. Instead, looking at accuracy only is insufficient to conclude. Although the authors gave some reasons about the usage of accuracy, I disagree with them all. My main concern is from the class imbalance in machine learning and the issue of *task diversity* here is very similar/related. In the case of class imbalance, simply using accuracy to measure the performance is not the right way. That is why there are many different metrics proposed for such cases. Here, the research tried to investigate the *task diversity* by using the accuracy, and hence, it is insufficient.
> >
> > > informatic modelling
> >
> > My apology for not making it clear. My main suggestion is to investigate the *task diversity* based on distribution or probabilistic modelling. In this case, we have a distribution of tasks where each task is a vector obtained from TASK2VEC. Could we use any tools in probabilistic modelling to conclude about such distribution?

---

### Official Review · Reviewer_3tgb · 2022-07-11

**Rating:** 4
**Confidence:** 4
**Soundness:** 2 fair
**Presentation:** 2 fair
**Contribution:** 2 fair

**Summary:**

This paper presents a novel metric which quantifies the diversity of few-shot learning benchmarks. The authors compare MAML with transfer learning w.r.t. multiple aspects on two few-shot learning benchmarks. Several empirical discoveries are presented including that transfer learning fails to outperform MAML when the intrinsic diversity is low for the evaluation benchmark.

**Questions:**

1. In the empirical evaluation, it seems that only classification task and ConvNet-based models are used. For making a concrete conclusion, more tasks and diverse architectures (e.g. transformer-based backbone) will be more persuasive. Otherwise, the scope of the conclusion needs to be narrowed down.
2. The definition of tasks is not clear in the caption of Table 1. Are the 500 few-shot learning tasks subsets from the original dataset? Do they have disjoint classes? Does changing the definition of these tasks results in different diversity coefficient?
3. Line 292, 396, the section number is missing.

**Limitations:**

The limitation of the work is discussed in Section 6. The potential negative societal impact is not discussed.

**Strengths And Weaknesses:**

Strengths:
- The paper is quite easy to follow.
- The paper compares MAML and transfer learning from an interesting perspective, the so-called "problem centric" approach.
- The idea of rethinking the diversity of datasets is important for analyzing models, and this paper makes a good case for it by conducting empirical evaluations.
- The diversity coefficient is intuitive.

Weakness:
- The empirical evaluation needs to be stronger.
- Some parts are not clear.
- Minor typos.
Please see questions for details.

---

> ### Author Response · Authors · 2022-08-01
> **The data centric diversity coefficient is a more effective way to count the number of truly different tasks compared to purely combinatorial computations**
>
> Hi 3tgb. Thanks for the direct, honest and concise review.
>
> Response to questions:
> In Figure 4, all 7 data sets that we trained were on a fully connected network. Thus, our experiments hold for fully connected nets for at least of 7 data sets with varying diversities in the 1D Gaussian setting. In addition, each pair of points in that figure is a data set, which means we evaluated 7 different data sets. We admit these tasks are synthetic, but their true diversity can be computed exactly using the Hellinger distance between distributions – which is impossible for any real task. We do admit it did not occur to us to do transformer-based experiments, since previous work claiming that transfer learning is better than MAML did not use transformers. Our work was mainly in response to that work (e.g. [8] Y. Tian, Y. Wang, D. Krishnan, J. B. Tenenbaum, and P. Isola, “Rethinking Few-Shot Image 325 Classification: a Good Embedding Is All You Need?,” 2020), but we think it’s an interesting idea to include transformer-based experiments in the future (e.g. ViT). But we don’t think it’s essential, since transformer based models have not unambiguously dominated the computer vision yet. We do want to express that to a point, it’s impossible to try all architectures and tried the most relevant ones for few-shot learning. However, we want to emphasize that instead we did an experiment with increasing model size given a fixed architecture (figure 3). Which adds a fascinating perspective to the problem, since currently there seems to be an emphasis on increasing model sizes. Perhaps one would expect meta-learning to help the most when the model is small, and we showed that this is not the case, even with tiny models. We argue this strengthens our hypothesis further in the most meaningful way, since it discards model size as an important factor for meta-learning – especially in the setting where one would expect it to be most influential where small models might benefit the most from meta-learning.
> We reference the definition of a task in section 3. It is the standard n-way k-shot classification task. We are happy to further clarify if needed. The 500 tasks is a subset of the 64 choose 5 ways to choose a task. Using MiniImagenet as an example, there are 64 labels, and we usually select 5 classes to form a task. The tasks are definitively unique, but given we used 500 – it is likely that some tasks share some labels. But this is not special to our work, this is how few-shot learning has been evaluated in the past. We do think the definition of a task must affect the value of diversity. We can easily provide experiments to see the effects of say, changing the shots, or the number of classes and see how the diversity coefficient changes if we want to explore that. However, few-shot learning is focused on few-shots (e.g. n=5) and that is how we computed our metric. We are unsure what insights it would provide to increase the shots since it wouldn’t be few shot anymore, but it's easy to try. Perhaps you will be interested in looking at figure 11 – where we plot the correlation of the Task2Vec diversity against the *true* Hellinger Diversity for a synthetic task. They correlated positively, and this is interesting because we can compute the true Hellinger distance between distributions in this case. Final comment on definition of a task, the diversity coefficient computes the difference between tasks, so it attempts to compute the *true way to count* how many truly different tasks there are. Contrast this with counting tasks using 64 choose 5. The latter doesn’t take the actual data into account in it’s counting, and thus is prone to over counting. We see diversity analogously to how VC-dimension attempts to count the true number of different models in a hypothesis class. We are happy to make these remarks  explicit in our paper & how we believe the current way to define a task is lacking.
> Oh, that is strange. We will for sure fix it (“Line 292, 396, the section number is missing.”) – should have been section K. For that, we used the MIT license.

---

### Official Review · Reviewer_vKuj · 2022-07-12

**Rating:** 4
**Confidence:** 3
**Soundness:** 2 fair
**Presentation:** 2 fair
**Contribution:** 2 fair

**Summary:**

This paper revisits the agreement of recent results of transfer learning methods outperforming meta-learning algorithms in few-shot learning domains, and claims that they are in fact not much different in performance particularly for a dataset with low diversity. To quantify the diversity, a new metric, called diversity coefficient, is proposed by leveraging the vectorized representation of each task obtained by Task2Vec. In empirical study, the authors' claim seems to be valid as the accuracy are pretty much similar to each other in between MAML and USL (i.e., a transfer learning method) using Cifar-FS and Mini-ImageNet, both of which have low diversity.

**Questions:**

1. The experimental results of Figure 2 is somewhat obvious. Are they intended to see models trained by MAML and USL are not the same in their weights?

2. Could you provide any insights on why diversity matters in the performance of MAML and USL?

3. Given a high diversity dataset, USL seems to be still better than MAML. Doesn't this imply that USL deals with a more challenging learning problem setting better than MAML?

4. How can we trust the proposed metric indeed well reflects the true diversity? Any experimental justification on that?

**Limitations:**

Potential negative societal impact is neither discussed nor applicable to this work.

**Strengths And Weaknesses:**

(Strengths)
1. The main claim of this paper sounds very important as it tries to break the common knowledge agreed by many recent works in few-shot learning.
2. Experiments are well designed in a way that the corresponding results properly justify the main claims and questions.

(Weaknesses)
1. Although the paper starts with ambitious insights, it does not properly give a meaningful conclusion in the end. There is no explanation on why low diversity leads to the similar performance of MAML and USL. Furthermore, when the diversity gets higher, it seems that the common knowledge is still correct as USL starts to outperform MAML. Thus, it would be much more interesting if the paper deeply investigates the hidden relationship between diversity and performance of both methodologies in few-shot learning. At the present form, the paper looks kind of an intermediate work on the way to its finalized version.

2. The proposed diversity metric is adequate, but its novelty is mostly coming from the corresponding preliminary work, i.e., Task2Vec. What is newly proposed in this work is to obtain the expected distance between tasks via their vectorized representations.

3. In terms of presentation quality, this write-up should be more improved and finalized. There are some typos and missing references throughout the paper, and most of figures are in low quality and hard to recognize.

---

> ### Author Response · Authors · 2022-08-01
> **We did provide a partial mechanistic explanation in the supplementary section but even if provided we respectfully disagree it's needed**
>
> We respectfully disagree that our results meaningful because they are not mechanistic. Our argument does the following:
> It avoids the current pattern in machine learning literature that appeals to vague intuition for definitions of “tasks” – especially, without quantifiable metrics. We provide a very concrete way to measure the properties of the data and see how those affect different algorithms.
> We provide a concrete way to go beyond collecting large data sets with vague claims of them being “diverse tasks”. We believe we move the field forward and away from appealing to intuition and instead use grounding data set creation on quantitative metrics.
> Most importantly, one can now relate the performance of different meta-learning algorithms to these quantitative properties of the data sets.
> To the best of our knowledge, this goes beyond what is being attempted right now as far as we know. In the most respectful manner and with a sincere effort to understand your perspective, why do you think our quantitative data-centric approach to meta-learning is not meaningful?
>
> We do want to make a comment on the mechanistic perspective. We actually do have some results on this in section K of the supplementary section.
> This is where perhaps its subjective. We (perhaps incorrectly) assume you think it’s important since it’s the first point you brought up. But during the research & writing process, we didn’t think so – to the point where we relegated it to the supplementary section. We are happy to fix that if the reviewers value it. However, we suspect that a complicated phenomenon – like separating meta-learning algorithms – is more realistically studied from an empirical study.
> However, our none exhaustive theoretical analysis did inspire this work.
>
> We’d like to address the second weakness point – the one that mentions that our use of Task2vec is insufficiently novel. The novelty is in the use of well-studied task embeddings to make non-trivial observations of meta-learning. In addition, our use of these metrics suggests a paradigm shift in the creation of benchmarks – the foundation of machine learning research.
>
> We committed to improving the writing and the figures for the final version.
>
> Answers to question:
> Q1 We don’t believe it is obvious that MAML and USL should learn to produce different (or similar) representations. From section K of the theory out, it was reasonable to hypothesize MAML and USL to learn similar representations under low diversity. In short, the experiments were meant to explore a. if they learned the same function since they have very similar accuracy and b. to test the hypothesis inspired by the theory
> Q2 Happy to clarify why diversity matters. The intuition is that the more diverse sampled tasks from a benchmark are, the more the algorithm has to learn to adapt/learn. Section K in the supplementary is an attempt to make that formal. Following that idea, if a task has low diversity, then the algorithm doesn’t need to be learn to learn. This is reflected in the extreme cases in section K: a. the decision function doesn’t need to change if task diversity is zero, or b. the task can be identified so the meta-learner has the capacity of adapting to produce the perfect decision function for that task. In the case where the diversity is low, we’d expect either method to be very similar – which is what we observed.
> Q3 We want to emphasize that our paper was meant to explore the low diversity regime in depth e.g. see the title of our paper. This is because the low diversity setting is already rich in empirical and theoretical analysis. If it is needed, please let us know explicitly and ideally with a reason so that we can act accordingly. Having said that, we are happy to respond. This question is excellent and nuanced. The high diversity regime can be in some cases easier. This is unintuitive, but in the high diversity regime there is more information to discriminate classes. However, in the low diversity setting, it is harder because the model has to work with less variation. This is especially clear in our synthetic experiments, where the tasks are harder to distinguish.
> Q4: We relied implicitly on the original justifications of Task2Vec. They showed that their vectors align with qualitative properties of tasks e.g. it correlates with taxonomic distance, and vectors cluster wrt taxonomies & semantics. We did not include a thorough recapitulation, but we are happy to provide it. Final comment on the relation of Task2vec diversity and “true diversity”. To do this, one needs a way to estimate the distance between the true task distributions. This is not usually available or it’s hard to estimate for high dimensional data. But in our synthetic experiments, we know exactly which Gaussians we used to generate tasks. Therefore, in figure 11 section H we explicitly correlate the Task2Vec diversity with the true diversity. We observed that they correlate well with an R-value of 0.990.

---

### Meta-Review · Area_Chair_NHcR · 2022-08-27

**Recommendation:** Reject
**Confidence:** Certain

**Metareview:**

The paper performs some empirical study between transfer learning and MAML (as a meta-learning method) through the lens of task diversity. When the task diversity is low, the authors claim that the performance of MAML and transfer learning methods are similar under a fair comparison (e.g. same architecture, optimizer etc). All reviewers are on a negative side for this paper due to weak experimental supports, poor write-up, weak novelty, etc, and the authors also fail to convince the reviewers through their rebuttal responses. Hence, AC cannot recommend acceptance at the current form. In particular, AC agrees that "the paper looks kind of an intermediate work on the way to its finalized version" and "I couldn't understand the clear takeaway or message from the paper except for certain empirical insights" pointed out by reviewers. AC thinks that this paper becomes much stronger if the authors can propose new better meta-learning benchmarks using the insights obtained through the authors' analysis.

**Award:**

No

---

### Decision · Program_Chairs · 2022-09-14

Reject